# Adaptive Multiscale Binary Expansion Tests for Independence

**Yang Yang** [1]  **Duo Zheng** [2]  **Sandeep Jain** [3]  **Kai Zhang** [4]  **Ping-Shou Zhong** [1]

## Abstract

This paper introduces a new family of adaptive, distribution-free independence tests for multivariate random vectors based on binary expansion coefficients, supported by a tractable asymptotic theory. Our first key contribution establishes a general equivalence between independence testing and testing cross-covariances among exponentially many binary expansion interaction coefficients, applicable to broad sample spaces and not limited to kernel-induced representations. While this exponential interaction structure makes naive construction and computation infeasible, we overcome this challenge by reformulating the proposed tests as a class of U-statistics and deriving an explicit kernel representation that enables scalable and efficient computation. Exploiting the multiscale nature of binary expansions, the proposed framework automatically adapts to unknown dependence structures by selectively truncating higher-order interactions, yielding both strong power and clear interpretability. To further enhance power and computational efficiency, we introduce an adaptive weighted aggregation procedure, termed wa-dCoBET, which combines a baseline Covariance Binary Expansion Test (CoBET) with a distance-measure–based CoBET. Extensive simulations and a real-data application demonstrate that wa-dCoBET consistently matches or outperforms HSIC and distance covariance, particularly in higher-dimensional and non-monotone settings, while maintaining accurate type I error control. A Python implementation of the proposed procedure is available at https://github.com/yyang3388/Cobet.

[1]Department of Mathematics, Statistics, and Computer Science, University of Illinois Chicago, Chicago, IL, USA [2]Amazon, Seattle, WA, USA [3]Department of Ophthalmology and Visual Science, University of Illinois Chicago, Chicago, IL, USA [4]Department of Statistics and Operations Research, University of North Carolina at Chapel Hill, Chapel Hill, NC, USA. Correspondence to: Ping-Shou Zhong <pszhong@uic.edu>.

*Proceedings of the 43$^{rd}$ International Conference on Machine Learning*, Seoul, South Korea. PMLR 306, 2026. Copyright 2026 by the author(s).

## 1. Introduction

Testing statistical independence between multivariate random variables is a fundamental problem in statistics, econometrics, and machine learning. Beyond classical applications such as feature selection, dimension reduction, and graphical modeling, independence tests have recently attracted increasing attention in areas including causal discovery (Dai et al., 2022) and large language models (Zhu et al., 2025).

Classical independence tests, such as the $\chi^2$ test, Fisher's exact test, and their extensions (Agresti, 2013), have been widely used to assess independence between univariate random variables. However, these methods are often less powerful and less reliable when applied to multivariate random variables. This limitation arises from small expected cell counts due to limited sample sizes, as well as the potentially large number of categories required to represent multivariate variables (Székely et al., 2007; Gretton et al., 2005). Besides contingency-table–based approaches, a broad class of independence tests measures the discrepancy between the joint distribution and the product of its marginals. Early examples include Hoeffding's test of independence (Hoeffding, 1948) and the related Blum–Kiefer–Rosenblatt coefficient (Blum et al., 1961). Modern extensions build on ranks, copulas, and empirical process theory to construct distribution-free tests that are invariant to monotone marginal transformations. Representative examples include consistent rank-based correlations (Bergsma & Dassios, 2014; Chatterjee, 2021), copula-based tests (Berg & Bakken, 2005; Kojadinovic & Yan, 2011; Genest et al., 2009), and the Heller-Heller-Gorfine test (Heller et al., 2016), which aggregates multiscale rank statistics to detect both monotone and non-monotone dependencies.

Another influential line of work quantifies dependence via the Kullback–Leibler divergence between the joint distribution and the product of the marginals, commonly known as mutual information. This perspective has motivated both $k$-nearest-neighbor estimators (Kraskov et al., 2004; Gao et al., 2017; Singh et al., 2003) and neural estimators (Belghazi et al., 2018; Poole et al., 2019) aimed at reducing estimation bias in moderate to high dimensions. More recently, learning- and projection-based paradigms have emerged, including classifier-based reductions (Lopez-Paz et al., 2017;

Muandet et al., 2017; Kim et al., 2020) and optimized or random projection methods (Zhu et al., 2017; Weihs et al., 2018). Despite these advances, nonparametric estimation of joint distributions for multivariate random variables remains a challenge for limited sample sizes.

The other two popular measures of independence without reliance on joint distribution functions are based on Hilbert–Schmidt Independence Criterion (HSIC) (Gretton et al., 2008) and distance correlation (Székely et al., 2007; Sejdinovic et al., 2013). The Hilbert–Schmidt Independence Criterion (HSIC) (Gretton et al., 2008) is the Hilbert–Schmidt norm of the cross-covariance operator, which is a generalization of covariance operator for random variables in a Euclidean space. It has been shown that HSIC=0 if and only if two random variables are independent provided that the kernel is c-universal (Gretton et al., 2005). C-universal means that all continuous functions can be approximated by certain functions in the reproducing kernel Hilbert space (RKHS) (Steinwart, 2012; Sriperumbudur et al., 2011). In other words, HSIC=0 is equivalent to independence if appropriate kernels are chosen. Distance correlation (Székely et al., 2007; Sejdinovic et al., 2013) quantifies statistical dependence through a weighted $L_2$-norm between the joint characteristic function and the product of the marginal characteristic functions. A key advantage of distance correlation lies in its carefully chosen weighting scheme, which enables efficient estimation of the resulting integrals even for multivariate random variables, without explicitly estimating either the joint or marginal characteristic functions. However, as pointed out by one referee, the weight function in distance correlation down weights the high-frequency components in the characteristic functions, which can potentially reduce power for detecting complex, non-linear, and local dependencies.

In this paper, we introduce a new independence measure by representing characteristic functions via binary expansions (Zhang, 2019; Zhang et al., 2021; Brown et al., 2025). The new independence measure translates an independence testing problem to a cross-covariance testing problem for binary expansion coefficients under a general setting, not limited to RKHS (hence not depending on the choices of kernels). Moreover, by analyzing the distance covariance measure (Székely et al., 2007) from a binary expansion perspective, we discover that the distance covariance measure is equivalent to a weighted quadratic distance between binary expansion coefficients, involving coefficients of binary expansion for all orders. The inclusion of all high-order binary expansion coefficients may introduce more noise than signal, potentially degrading the power of the distance correlation test. The proposed measure has the flexibility to choose resolution so that we can retain only lower-order coefficients to strike a better balance between signal and noise. In addition, the proposed measure substantially improves

interpretability by identifying the components that exhibit dependence when the null hypothesis of independence is rejected (see numerical studies for an illustration).

Based on the proposed independence measure, we develop a unified framework for testing weighted cross-covariances between binary expansion interaction coefficients. A direct construction is computationally infeasible, as the number of such interaction coefficients grows exponentially with the dimension and expansion depth. To address this challenge, we introduce a family of U-statistic–based test statistics and derive equivalent kernel representations that dramatically reduce the computational complexity from $O(2^{dK})$ to $O(dK)$, enabling scalable and efficient implementation, where $d$ is the order of the dimension of random vectors and $K$ is a chosen resolution for binary expansion.

Through the use of weight matrices, the proposed methodology offers additional flexibility and the potential to boost testing power. We investigate two special choices of weights. The first is the identity weight, which yields a simple baseline test termed the Covariance Binary Expansion Test (CoBET). The second is a weight function derived from distance correlation, leading to dCoBET, which incorporates positive semidefinite feature weights inspired by distance-covariance constructions and establishes a direct link between binary expansions and classical distance-based measures. To adaptively combine complementary weighting schemes, we further introduce wa-dCoBET, which aggregates identity and distance-based weights via sample splitting and foldwise signal-to-noise ratio voting (He et al., 2023), while retaining a tractable asymptotic null distribution that allows calibration without permutation.

Simulation studies based on multivariate copulas demonstrate that the proposed methods achieve competitive or superior power across a wide range of linear and nonlinear dependence structures, while remaining computationally efficient and theoretically transparent.

The remainder of the paper is organized as follows. Section 2 introduce the binary-expansion based independence measure, the CoBET, dCoBET, and wa-dCoBET statistics and their kernel representations and asymptotic distributions. Section 3 presents simulation results, Section 4 illustrates the method on real data, and Section 5 concludes. Additional details and technical proofs are provided in the Appendix.

## 2. Independence Measure and Testing Procedure

Let $\{(X_i, Y_i)\}_{i=1}^n$ be i.i.d. copies of $(X, Y) \in \mathbb{R}^p \times \mathbb{R}^q$. We test

$$H_0 : X \perp Y \quad \text{vs} \quad H_1 : X \not\perp Y,$$

where $X \perp Y$ means $X$ is independent of $Y$ and $X \not\perp Y$ means $X$ is not independent of $Y$. This section will first

introduce a new binary expansion based independence measure, and then introduce the proposed $U$–statistic for testing cross-covariance and derive a computationally efficient and scalable kernel representation of the proposed statistics, and its asymptotic theory.

## 2.1. A binary expansion based independence measure

To simplify our notations, we perform componentwise probability–integral transforms to $X_i$ and $Y_i$ by $\tilde{U}_i^{(r)} = F_{X^{(r)}}(X_i^{(r)})$ and $\tilde{V}_i^{(s)} = F_{Y^{(s)}}(Y_i^{(s)})$ to obtain random vectors $\tilde{U}_i$ and $\tilde{V}_i$ where

$$\tilde{U}_i = (\tilde{U}_i^{(1)}, \ldots, \tilde{U}_i^{(p)}), \qquad \tilde{V}_i = (\tilde{V}_i^{(1)}, \ldots, \tilde{V}_i^{(q)})$$

for pairs of samples $(\tilde{U}_i, \tilde{V}_i)$ where $i = 1, \cdots, n$ that are independent copies of $(\tilde{U}, \tilde{V})$. For each transformed coordinate $\tilde{U}_i^{(r)} \in [0,1]$ and $\tilde{V}_i^{(s)} \in [0,1]$, define $U_i^{(r)} = 2\tilde{U}_i^{(r)} - 1$ and $V_i^{(s)} = 2\tilde{V}_i^{(s)} - 1$. It is clear that

$$X \perp Y \iff \tilde{U} \perp \tilde{V} \iff U \perp V.$$

The random vectors $U_i$ and $V_i$ have uniform marginals on $[-1, 1]$ and satisfy $X_i \perp Y_i \iff U_i \perp V_i$. Independence between $U$ and $V$ admits the classical characterization via characteristic functions:

$$U \perp V \iff \varphi_{UV}(t, s) = \varphi_U(t)\varphi_V(s), \quad (t, s) \in \mathbb{R}^p \times \mathbb{R}^q.$$

where $\varphi_{UV}$ denotes the joint characteristic function and $\varphi_U$, $\varphi_V$ denote the marginal characteristic functions.

Consider the binary expansion for $U$ and $V$,

$$U_i^{(r)} = \sum_{k=1}^{\infty} 2^{-k} A_{k,i}^{(r)}, \quad V_i^{(s)} = \sum_{j=1}^{\infty} 2^{-j} B_{j,i}^{(s)},$$

where $A_{k,i}^{(r)}, B_{j,i}^{(s)}$ are random coefficients that are Bernoulli distributed taking values in $\{-1, +1\}$. We truncate the binary expansions at a finite depth $K \in \mathbb{N}$, which controls the *resolution* of the representation. For notational simplicity, we assume the resolution for all the components are $K$. By approximation theory, we may ignore the remainder term beyond depth $K$; the resulting approximation error vanishes as $K \to \infty$.

Applying Euler's identity $e^{\jmath z} = \cos z + \jmath \sin z$ yields

$$e^{\jmath t_r U_i^{(r)}} = \prod_{k=1}^{K} \left\{ \cos\left(\frac{t_r}{2^k}\right) + \jmath A_{k,i}^{(r)} \sin\left(\frac{t_r}{2^k}\right) \right\}, \jmath = \sqrt{-1}.$$

Taking expectations gives a polynomial–trigonometric representation of the marginal characteristic function

$$\varphi_{U^{(r)}}(t_r) = \sum_{a \in \{0,1\}^K} \psi_a(t_r) \, \mathbb{E}\left( \prod_{k=1}^{K} (A_k^{(r)})^{a_k} \right),$$

where each coefficient $\psi_a(t_r)$ is formed from products of $\cos(t_r/2^k)$ and $\sin(t_r/2^k)$. The same construction applies to each coordinate of $V$.

For a fixed depth $K$, let $I_r$ be the collection of all the nonempty subsets of $\{1, \ldots, K\}$. Define the multi-index set

$$\mathcal{I}_p^* = \left\{ \begin{array}{l} I = (I_1, \ldots, I_p) : \\ I_r \subset \{1, \ldots, K\}, \ I_r \neq \emptyset, \ 1 \le r \le p \end{array} \right\},$$

and analogously define $\mathcal{I}_q^*$. Let $|I_r|$ be the cardinality of the set $I_r$.

**Theorem 2.1.** *Let $U = (U^{(1)}, \ldots, U^{(p)}) \in [-1, 1]^p$ and $V = (V^{(1)}, \ldots, V^{(q)}) \in [-1, 1]^q$ be the componentwise probability–integral transforms of $X$ and $Y$, and fix a truncation depth $K \in \mathbb{N}$. For $I \in \mathcal{I}_p^*$, write $|I| = \sum_{r=1}^{p} |I_r|$ and define*

$$\Pi_I(t) = \prod_{r=1}^{p} \left( \prod_{l \in I_r} \sin \frac{t_r}{2^l} \prod_{l \notin I_r} \cos \frac{t_r}{2^l} \right), \qquad t \in \mathbb{R}^p,$$

*with $\Pi_J(s)$ defined analogously for $J \in \mathcal{I}_q^*$. Then*

$$\varphi_{UV}(t, s) - \varphi_U(t)\varphi_V(s) = \sum_{I \in \mathcal{I}_p^*} \sum_{J \in \mathcal{I}_q^*} \jmath^{|I|+|J|} m_{I,J}$$
$$\times \Pi_I(t) \, \Pi_J(s).$$

*where $I = (I_1, \ldots, I_p) \in \mathcal{I}_p^*$ and $J = (J_1, \ldots, J_q) \in \mathcal{I}_q^*$, with $I_r, J_s \subseteq \{1, \ldots, K\}$, and*

$$m_{I,J} = \mathbb{E}(\mathcal{A}_{i,I}\mathcal{B}_{i,J}) - \mathbb{E}(\mathcal{A}_{i,I})\,\mathbb{E}(\mathcal{B}_{i,J}),$$
$$\mathcal{A}_{i,I} = \prod_{r=1}^{p} \prod_{k \in I_r} A_{k,i}^{(r)}, \qquad \mathcal{B}_{i,J} = \prod_{s=1}^{q} \prod_{j \in J_s} B_{j,i}^{(s)}.$$

The proof of Theorem 2.1 is included in Appendix B.1. The expansion in Theorem 2.1 is related to the universal binary expansion approximation of characteristic functions developed in (Zhang, 2019; Zhang et al., 2021). Compared to these results, Theorem 2.1 can be viewed as a corresponding decomposition of independence: The essential difference $\varphi_{UV}(t, s) - \varphi_U(t)\varphi_V(s)$ is expressed as a finite linear combination of cross-covariance coefficients $m_{I,J} = \text{Cov}(\mathcal{A}_{i,I}, \mathcal{B}_{i,J})$ with basis functions $\Pi_I(t)\Pi_J(s)$.

This covariance form is crucial in the multivariate setting, since within-vector dependence generally implies nonzero marginal interaction means, and therefore any form of dependence is naturally characterized by all cross-covariances $\{m_{I,J}\}$. As a direct consequence, testing $U \perp V$ reduces to testing whether the collection $\{m_{I,J}\}$ vanishes. This translates the independent testing problem into a covariance test problem, which is an easier problem. More specifically, define the feature vectors

$$\vec{A}_i = (\mathcal{A}_{i,I})_{I \in \mathcal{I}_p^*} \in \mathbb{R}^{d_A}, \quad \vec{B}_i = (\mathcal{B}_{i,J})_{J \in \mathcal{I}_q^*} \in \mathbb{R}^{d_B},$$

with $d_A = 2^{pK} - 1$, $d_B = 2^{qK} - 1$ and define the cross-covariance matrix

$$\Sigma_{AB} = \mathrm{Cov}(\vec{A}_i, \vec{B}_i), \qquad \Sigma_{BA} = \Sigma_{AB}^\top.$$

A specific example of $\mathcal{I}_p^*$, and $\vec{A}_i$ is given in the Appendix B.7

The results in Theorem 2.1 imply that $U$ and $V$ are independent if and only if $\Sigma_{AB} = 0$. The binary expansion feature vectors $\vec{A}$ and $\vec{B}$ have dimensions $d_A = 2^{pK} - 1$ and $d_B = 2^{qK} - 1$, which grows exponentially fast as $K$ and $p$ or $q$ grow. Fortunately, this only serves as a theoretical expression for easy understanding. The proposed procedures do NOT construct it explicitly and hence it overcomes the computation issues of using $\vec{A}$ and $\vec{B}$ directly. See Section 2.2 for details.

More generally, define a measure of independence between joint characteristic function and marginal characteristic functions

$$\mathcal{V}^2(U, V) = \int_{\mathbb{R}^p} \int_{\mathbb{R}^q} |\varphi_{UV}(t,s) - \varphi_U(t)\varphi_V(s)|^2 w^2(t,s) dt ds,$$

where $w^2(t, s)$ is a non-negative weight function. This weight measure includes distance covariance measure as a special case. We write $\mathcal{V}^2(U, V)$ as $\mathcal{V}_W^2(U, V)$ if the weight $w(t, s) = w_0(t) w_1(s)$ is applied and write $\mathcal{V}^2(U, V)$ as $\mathcal{V}_D^2(U, V)$ if the distance-covariance weight $w^2(t, s) = 1/\{\|t\|^{p+1}\|s\|^{q+1}\}$ (Székely et al., 2007) is used. Applying Theorem 2.1, we have

$$\mathcal{V}_{W,K}^2(U, V) = \sum_{I, I' \in \mathcal{I}_p} \sum_{J, J' \in \mathcal{I}_q} m_{I,J} W_{I,I'}^{(p)} W_{J,J'}^{(q)} m_{I',J'}$$
$$= \mathrm{tr}(\Sigma_{AB} W_B \Sigma_{BA} W_A),$$

where the weight matrix $W_A = (W_{I,I'}^{(p)})_{I,I' \in \mathcal{I}_p^*} \in \mathbb{R}^{d_A \times d_A}$ has entries

$$W_{I,I'}^{(p)} = \int_{\mathbb{R}^p} \Pi_I(t) \Pi_{I'}(t) w_0^2(t) dt.$$

The weight matrix $W_B \in \mathbb{R}^{d_B \times d_B}$ is defined analogously to $W_A$ by replacing $p, \mathcal{I}_p^\star, \Pi_I(t), w_0(t)$ with $q, \mathcal{I}_q^\star, \Pi_J(s)$ and $w_1(s)$.

If one chooses the distance-covariance weight (Székely et al., 2007), the resulting population distance covariance $\mathcal{V}_{D,K}^2(U, V)$ admits

$$\mathcal{V}_{D,K}^2(U, V) = \mathrm{tr}\left(\Sigma_{AB} K^{(q)} \Sigma_{BA} K^{(p)}\right),$$

where for $d \in \{p, q\}$,

$$K_{I,I'}^{(d)} = \frac{1}{c_d} \int_{\mathbb{R}^d} \frac{\Pi_I(t)\Pi_{I'}(t)}{\|t\|^{d+1}} dt, \qquad I, I' \in \mathcal{I}_d^\star.$$

where $c_d$ is a constant. Both $K^{(p)}$ and $K^{(q)}$ are symmetric positive semidefinite with closed-form expressions depending only on $(p, K)$ and $(q, K)$, respectively; no numerical integration is required (Appendix B.3). When $K$ is large, $\mathcal{V}_{D,K}^2(U, V)$ converges to its limit distance covariance measure $\mathcal{V}_D^2(U, V)$.

## 2.2. Covariance Binary Expansion Tests

Theorem 2.1 shows that departures from independence between $U$ and $V$ are fully characterized by the collection of covariances $\{m_{I,J}\}_{I \in \mathcal{I}_p^\star, J \in \mathcal{I}_q^\star}$. Equivalently, independence is characterized by the vanishing of the cross–covariance matrix or the distance measures $\mathcal{V}_W^2(U, V)$ defined in Section 2.1.

**CoBET.** The simplest dependence measure aggregates all binary–expansion interactions with identity weights $w_0(t) = 1$ and $w_1(s) = 1$ in $\mathcal{V}_{W,K}^2(U, V)$ defined in Section 2.1, write it as $\mathcal{V}_{I,K}^2(U, V)$. More specifically, define the population functional

$$\mathcal{V}_{I,K}^2(U, V) = \mathrm{tr}(\Sigma_{AB}\Sigma_{BA}) := \|\Sigma_{AB}\|_F^2.$$

The corresponding sample statistic to estimate $\mathcal{V}_I^2(U, V)$ is constructed as an unbiased $U$–statistic, defined below $T_n^{(W)}$ with $W_A$ and $W_B$ identity matrices, and is referred to as the *Covariance Binary Expansion Test (CoBET)*.

**dCoBET.** More generally, let $W_A$ and $W_B$ be matrices defined in Section 2.1 of sizes, respectively, $d_A \times d_A$ and $d_B \times d_B$. The corresponding population quadratic functional

$$\mathcal{V}_{W,K}^2(U, V) = \mathrm{tr}\left(\Sigma_{AB} W_B \Sigma_{BA} W_A\right).$$

By Theorem 2.1, $\mathcal{V}_W^2(U, V) = 0 \iff H_0 : U \perp V$ for any non-negative definite matrices $W_A, W_B$ defined in Section 2.1. Choosing distance-covariance motivated weights (e.g., $W_A = K^{(p)}$, $W_B = K^{(q)}$) yields the *dCoBET* family. Although dCoBET and distance covariance share similar weight function, distance covariance is only equivalent to dCoBET when $K$ is large enough, but dCoBET has the flexibility to truncate the binary expansion to remove independent components with high-variant.

**Unbiased $U$–statistics for CoBET and dCoBET.** Given i.i.d. samples $\{(U_i, V_i)\}_{i=1}^n$, and recall that the binary expansion interaction coefficient vectors $\vec{A}_i = (A_{i,I})_{I \in \mathcal{I}_p^\star}$ and $\vec{B}_i = (B_{i,J})_{J \in \mathcal{I}_q^\star}$ in Section 2.1. Define

$$T_n^{(W)} = T_{1n}^{(W)} - 2T_{2n}^{(W)} + T_{3n}^{(W)},$$

with

$$T_{1n}^{(W)} = \frac{1}{n(n-1)} \sum_{i \neq j} (\vec{A}_i^\top W_A \vec{A}_j)(\vec{B}_j^\top W_B \vec{B}_i),$$

$$T_{2n}^{(W)} = \frac{1}{\binom{n}{3}} \sum_{i \neq j \neq k} (\vec{A}_i^\top W_A \vec{A}_j)(\vec{B}_j^\top W_B \vec{B}_k),$$

$$T_{3n}^{(W)} = \frac{1}{\binom{n}{4}} \sum_{i \neq j \neq k \neq \ell} (\vec{A}_i^\top W_A \vec{A}_j)(\vec{B}_k^\top W_B \vec{B}_\ell),$$

where $\binom{n}{k}$ is a binomial coefficient with $k \leq n$. The second and third terms $T_{2n}^{(W)}$ and $T_{3n}^{(W)}$ are used to cancel the non-zeros means in the vectors $\vec{A}_i$ and $\vec{B}_i$ so that (see the proof in the Appendix B.2)

$$\mathbb{E}\left\{ T_n^{(W)} \right\} = \mathcal{V}_{W,K}^2(U, V).$$

Although the second and third terms $T_{2n}^{(W)}$ and $T_{3n}^{(W)}$ involve summations over $O(n^3)$ and $O(n^4)$ terms, respectively, they can be computed at the order of $O(n^2)$ (See the details in the Appendix B.3). The construction of $T_n^{(W)}$ is motivated by (Chen et al., 2010) but an essential and crucial difference is we do NOT have $\vec{A}_i$ and $\vec{B}_i$ available, because $\vec{A}_i$ and $\vec{B}_i$ have dimensions $d_A = 2^{pK} - 1$ and $d_B = 2^{qK} - 1$, which grows exponentially fast as $K$ and $p$ or $q$ grows and hence constructing them explicitly is computationally infeasible. Thus, the format in $T_n^{(W)}$ only serves for analyzing its theoretical property.

To compute the proposed CoBET and dCoBET statistics, we note that $T_n^{(W)}$ depends on the terms such as $\vec{A}_i^\top \vec{A}_j$ (when $W_A = I_{d_A}$) and $\vec{A}_i^\top W_A \vec{A}_j$. The following theorem shows how to compute them at the order of $O(pK)$.

**Theorem 2.2.** *For any $i \neq j$,*

$$\vec{A}_i^\top \vec{A}_j = \prod_{r=1}^{p} \prod_{k=1}^{K} \left(1 + A_{k,i}^{(r)} A_{k,j}^{(r)}\right) - 1. \qquad (1)$$

*Moreover,*

$$\vec{A}_i^\top W_A \vec{A}_j = \int_{\mathbb{R}^p} C^2(t) \left\{ \prod_{r=1}^{p} \prod_{k=1}^{K} \left(1 + A_{k,i}^{(r)} \pi_k^{(r)}(t_r)\right) - 1 \right\}$$

$$\times \left\{ \prod_{r=1}^{p} \prod_{k=1}^{K} \left(1 + A_{k,j}^{(r)} \pi_k^{(r)}(t_r)\right) - 1 \right\} w_0^2(t)\, dt,$$

*where $C^2(t) = \prod_{r=1}^{p} \prod_{k=1}^{K} \cos^2(t_r/2^k)$ and $\pi_k^{(r)}(t_r) = \tan(t_r/2^k)$.*

Proof of Theorem 2.2 is given in Appendix B.4. Equation (1) shows that the CoBET kernel $\vec{A}_i^\top \vec{A}_j$ can be evaluated in $O(pK)$ time using only pairwise bit products, despite the exponentially growing ambient dimension of $\vec{A}_i$. Similarly,

evaluating $\vec{A}_i^\top W_A \vec{A}_j$ via the representation in Theorem 2.2 reduces to forming the $O(pK)$ products inside the braces for each integration point $t$, rather than enumerating the $2^{pK} - 1$ interaction coordinates. Analogous results hold for $\vec{B}_i^\top \vec{B}_j$ (when $W_B = I_{d_B}$) and for $\vec{B}_i^\top W_B \vec{B}_j$. In particular, if $w_0^2(t) = \prod_{r=1}^{p} w_{0r}^2(t_r)$, we can further write $\vec{A}_i^\top W_A \vec{A}_j$ as a product of marginal integrations, which can be computed even more efficiently. See the Appendix B.4 for the details.

The following Theorem 2.3 provides the leading order variance of the test statistic $T_n^{(W)}$.

**Theorem 2.3.** *Under the null hypothesis $H_0 : U \perp V$, the leading order variance of $T_n^{(W)}$ is*

$$\mathrm{Var}\left(T_n^{(W)}\right) = \frac{2}{n(n-1)} \|W_A \Sigma_A\|_F^2 \|W_B \Sigma_B\|_F^2 \{1 + o(1)\}. \qquad (2)$$

For CoBET ($W_A = I_{d_A}, W_B = I_{d_B}$), (2) reduces to $\mathrm{Var}(T_n^{(W)}) = 2\|\Sigma_A\|_F^2 \|\Sigma_B\|_F^2 / \{n(n-1)\}$. To estimate the leading order variance, a ratio-consistent estimator can be obtained by using U-statistics that are similar to the construction of the test statistic $T_n^{(W)}$. More specifically, we can estimate $\|W_A \Sigma_A\|_F^2$ by

$$\|\widehat{W_A \Sigma_A}\|_F^2 = \frac{1}{n(n-1)} \sum_{i \neq j} (\vec{A}_i^\top W_A \vec{A}_j)^2$$

$$- \frac{2}{\binom{n}{3}} \sum_{i \neq j \neq k} (\vec{A}_i^\top W_A \vec{A}_j)(\vec{A}_j^\top W_A \vec{A}_k)$$

$$+ \frac{1}{\binom{n}{4}} \sum_{i \neq j \neq k \neq \ell} (\vec{A}_i^\top W_A \vec{A}_j)(\vec{A}_k^\top W_A \vec{A}_\ell).$$

An estimator of $\|W_B \Sigma_B\|_F^2$ can be constructed similarly. Note that both estimators $\|\widehat{W_A \Sigma_A}\|_F^2$ and $\|\widehat{W_B \Sigma_B}\|_F^2$ depend only on the terms like $\vec{A}_i^\top \vec{A}_j$ and $\vec{A}_i^\top W_A \vec{A}_j$, which can be computed efficiently using Theorem 2.2 without expanding $\vec{A}_i$ and $\vec{B}_i$ explicitly.

The following Theorem 2.4 establishes the limiting distribution of dCoBET and CoBET statistics under the null hypothesis.

**Theorem 2.4.** *Assume $\{(U_i, V_i)\}_{i=1}^n$ are i.i.d. random vector pairs, and $W_A, W_B$ are deterministic symmetric matrices. Assume the binary expansion depth $K = K(n) \to \infty$ as $n \to \infty$. Let $\widehat{\mathrm{Var}}(T_n^{(W)})$ be a ratio-consistent estimator of $\mathrm{Var}(T_n^{(W)})$. If $tr\{(\Sigma_A W_A)^4\} = o\left[tr^2\{(\Sigma_A W_A)^2\}\right]$ and $tr\{(\Sigma_B W_B)^4\} = o\left[tr^2\{(\Sigma_B W_B)^2\}\right]$, under $H_0 : U \perp V$, then*

$$Z_n^{(W)} := \frac{T_n^{(W)}}{\sqrt{\widehat{\mathrm{Var}}(T_n^{(W)})}} \stackrel{d}{\Rightarrow} \mathcal{N}(0, 1),$$

*where $\stackrel{d}{\Rightarrow}$ denotes converging in distribution.*

## 2.3. Weighted Aggregated dCoBET (wa-dCoBET)

CoBET and dCoBET are constructed using different weighting matrices. Although both weighting matrices maintain the type I error under the null, the power of tests may depend on the choice of weighting matrices due to the underlying dependence structure. To make our procedure adaptive to weight matrices, we introduce a *weighted aggregated dCoBET* (wa-dCoBET) statistic, which automatically selects and blends multiple weighting schemes using a data-driven signal-to-noise ratio (SNR) criterion (He et al., 2023).

We consider two positive weight pairs $W^{(\text{id})} = \left( I_{d_A}, I_{d_B} \right)$, $W^{(D)} = \left( K^{(p)}, K^{(q)} \right)$, corresponding to the identity weighted CoBET choice and the distance-covariance-weighted dCoBET choice, respectively. See Section 2.2 for details. For chosen $W = (W_A, W_B)$, let $T_n^{(W)}$ and $Z_n^{(W)}$ denote the corresponding $U$–statistic and its standardized version defined in Section 2.2, with variance estimated using estimator given in Section 2.2. More specifically, $Z_n^{(\text{id})}$ is the standardized statistic with identity weight $W = W^{(id)}$ (CoBET) and $Z_n^{(D)}$ is the standardized statistic with distance covariance motivated weight $W = W^{(D)}$ (dCoBET).

**Foldwise SNR voting.** We split the data into $M$ folds (e.g., $M = 10$) and compute foldwise standardized statistics $\text{SNR}_{n,m}^{(\text{id})}$ and $\text{SNR}_{n,m}^{(D)}$. Each fold votes for the weighting with the larger SNR, yielding empirical weights $\hat{w}_{\text{id}} = \frac{1}{M} \sum_{m=1}^{M} \mathbf{1}\{\text{SNR}_{n,m}^{(\text{id})} > \text{SNR}_{n,m}^{(D)}\}$, with $\hat{w}_D = 1 - w_{\text{id}}$.

where $\text{SNR}_{n,m}^{(\text{id})}$ and $\text{SNR}_{n,m}^{(D)}$ denote the estimated signal–to–noise ratios based on $Z_{n,m}^{(\text{id})}$ and $Z_{n,m}^{(D)}$, respectively.

The empirical vote proportions are used to form the aggregated weights $\hat{W}_A^{(\text{agg})} = \hat{w}_{\text{id}} I_{d_A} + \hat{w}_D K^{(p)}$ and $\hat{W}_B^{(\text{agg})} = \hat{w}_{\text{id}} I_{d_B} + \hat{w}_D K^{(q)}$. Evaluating the statistic with these aggregated weights yields the final standardized statistic $Z_n^{(\text{agg})}$.

The procedure strictly generalizes CoBET and dCoBET: setting $w_{\text{id}} = 1$ recovers CoBET, while $w_D = 1$ recovers dCoBET with $(K^{(p)}, K^{(q)})$. Foldwise SNR voting provides adaptivity while mitigating overfitting. When the number of folds $M$ is large, the estimated $\hat{w}_{id}$ and $\hat{w}_D$ converge (in probability) to their limits $w_{id}$ and $w_D$. Hence, the aggregated weight $\hat{W}_A^{(\text{agg})}$ converges in probability to its limit $W_A^{(\text{agg})}$. Then, applying Theorem 2.4 and the Slutsky Theorem, under $H_0 : U \perp V$ and $K = K(n) \to \infty$, $Z_n^{(\text{agg})} \overset{d}{\Rightarrow} \mathcal{N}(0, 1)$ as $n \to \infty$, yielding an asymptotically level-$\alpha$ test. For details of wa-dCoBET, please refer to Algorithm 1.

The resolution hyperparameter $K$ could be selected in a data-driven manor by maximizing the SNR of the power function. To provide practical guidelines and investigate the robustness of the proposed method to $K$, we conducted a

sensitivity analysis in Appendix (Section A.7).

If the null hypothesis is rejected, the source of dependence can be further investigated by adding an additional step. Specifically, when the dimensions of the random vectors are moderate, the global statistic can be decomposed into a sum of marginal statistics and their interactions. In higher-dimensional settings, such a decomposition may not be feasible. Instead, one can assess the contribution of each dimension to the aggregated adaptive statistic by removing one dimension at a time and ranking their relative importance. This ranking would provide useful insights into the underlying dependence structure.

---

**Algorithm 1** wa-dCoBET via 10-fold SNR voting

---

**Require:** $\{(U_i, V_i)\}_{i=1}^n$; depth $K$; $M = 10$ folds; $W^{(\text{id})} = (I_{d_A}, I_{d_B})$, $W^{(D)} = (K^{(p)}, K^{(q)})$; level $\alpha$.

**Ensure:** $Z_n^{(\text{agg})}$ and $\mathbb{1}\{Z_n^{(\text{agg})} > z_{1-\alpha}\}$.

  Precompute $\vec{A}_i^\top \vec{A}_j$, $\vec{A}_i^\top W^{(D)} \vec{A}_j$, $\vec{B}_i^\top \vec{B}_j$, $\vec{B}_i^\top W^{(D)} \vec{B}_j$ for $i \neq j$ (Theorem 2.2).

  Partition $\{1, \ldots, n\}$ into folds $F_1, \ldots, F_M$.

  **for** $m = 1$ to $M$ **do**

    Compute $\text{SNR}_{n,m}^{(\text{id})}$ and $\text{SNR}_{n,m}^{(D)}$ on $F_m$ via

    $$\text{SNR}_{n,m}^{(W)} := T_n^{(W)}(F_m) / \sqrt{\widehat{\text{Var}}\left( \tilde{T}_n^{(W)} \right)(F_m)}.$$

    $S_m \leftarrow \arg\max\{\text{SNR}_m^{(\text{id})}, \text{SNR}_m^{(D)}\}.$

  **end for**

  $\hat{w}_{\text{id}} \leftarrow \frac{1}{M} \sum_{m=1}^{M} \mathbb{1}\{S_m = \text{id}\}, \quad \hat{w}_D \leftarrow 1 - w_{\text{id}}.$

  $\hat{W}_A^{(\text{agg})} \leftarrow \hat{w}_{\text{id}} I_{d_A} + \hat{w}_D K^{(p)}, \; \hat{W}_B^{(\text{agg})} \leftarrow \hat{w}_{\text{id}} I_{d_B} + \hat{w}_D K^{(q)}.$

  Compute $T_n^{(\text{agg})} := T_n^{(W^{(\text{agg})})}$ and $\widehat{\text{Var}}(T_n^{(\text{agg})})$ on the full sample.

  $Z_n^{(\text{agg})} \leftarrow T_n^{(\text{agg})} / \sqrt{\widehat{\text{Var}}(T_n^{(\text{agg})})}.$

  **Reject** $H_0$ iff $Z_n^{(\text{agg})} > z_{1-\alpha}$.

---

## 3. Numerical Studies

We evaluate the finite-sample performance of CoBET, dCoBET, and wa-dCoBET in terms of empirical size and power, and compare them with several representative baselines, including HSIC (Gretton et al., 2012), distance covariance (dCov) (Székely et al., 2007), the randomized dependence coefficient (RDC) (Lopez-Paz et al., 2017), Chatterjee's rank-based correlation with mean aggregation across coordinate pairs (Xi-mean) (Chatterjee, 2021), and the multi-scale Fisher independence test (MultiFIT) (Gorsky & Ma, 2022). The experiments vary sample size, dimension, dependence strength, and the form of the relationship between $X$ and $Y$.

### 3.1. Data-generating mechanisms

We evaluate the proposed methods using synthetic data generated from a copula-based model that allows precise control

over both marginal dependence within covariates and the strength of dependence between the variables of interest. Specifically, for each observation $i = 1, \ldots, n$, we generate latent random vectors $(\tilde{U}_i, \tilde{V}_i) \in (0,1)^d \times (0,1)^d$, which are then mapped to observable variables $(X_i, Y_i) \in \mathbb{R}^p \times \mathbb{R}^q$ via coordinate-wise transformations.

**Copula-based covariates.** Dependence among covariates is induced through a $d$-dimensional Clayton copula with parameter $\theta \geq 0$. The Clayton copula allows us to vary the strength of lower-tail dependence while maintaining uniform marginals. Specifically, for each $i$ and $\theta > 0$, we generate

$$S_i \sim \text{Gamma}\left(\frac{1}{\theta}, 1\right), \qquad E_i^{(r)} \sim \text{Exp}(1), \quad r = 1, \ldots, d,$$

independently, and define

$$\tilde{U}_i^{(r)} = \left(1 + E_i^{(r)}/S_i\right)^{-1/\theta}, \qquad r = 1, \ldots, d.$$

The resulting vector $\tilde{U}_i = (\tilde{U}_i^{(1)}, \ldots, \tilde{U}_i^{(d)})$ follows a $d$-dimensional Clayton copula with uniform marginals. When $\theta = 0$, we instead generate $\tilde{U}_i^{(r)} \sim \text{Unif}(0,1)$ independently, corresponding to the independence copula.

Independently of $\tilde{U}_i$, we generate

$$\tilde{V}_i \sim \text{Unif}(0,1)^d,$$

which serves as an additional source of randomness in the response construction.

**Dependence between $X$ and $Y$.** Dependence between the variables of interest is controlled by a scalar parameter $b \geq 0$. When $b = 0$, the response depends only on $\tilde{V}_i$, yielding independence between $X$ and $Y$. When $b > 0$, both $X$ and $Y$ depend on $\tilde{U}_i$, inducing dependence between $X$ and $Y$ through shared latent structure. This design allows us to smoothly vary the signal strength while keeping the marginal distributions fixed.

**Coordinate-wise transformations.** Given $(\tilde{U}_i, \tilde{V}_i)$, we construct $(X_i, Y_i) \in \mathbb{R}^p \times \mathbb{R}^q$ with $p = q = d$ by applying one of the following transformations independently across coordinates. Let $Z_i^{(r)} = \Phi^{-1}(\tilde{U}_i^{(r)})$ and $b \geq 0$.

1. **Trigonometric (Trig):** $X_i^{(r)} = \sin(Z_i^{(r)}), \quad Y_i^{(r)} = \cos(bX_i^{(r)} + \tilde{V}_i^{(r)}).$

2. **Exponential–quadratic (ExpQuad):**

$$X_i^{(r)} = \exp\left\{-(Z_i^{(r)})^2\right\},$$

$$Y_i^{(r)} = \exp\left\{-b\left(X_i^{(r)} - 1\right)^2 + \tilde{V}_i^{(r)}\right\}.$$

3. **Linear:** $X_i^{(r)} = \tilde{U}_i^{(r)}, \quad Y_i^{(r)} = bX_i^{(r)} + \tilde{V}_i^{(r)}.$

4. **Log–quadratic (LogQuad):**

$$X_i^{(r)} = \frac{\log\{1 + (Z_i^{(r)})^2\}}{1 + \log\{1 + (Z_i^{(r)})^2\}},$$

$$Y_i^{(r)} = \cos(bX_i^{(r)} + \tilde{V}_i^{(r)}) \exp\{-b(X_i^{(r)} - 0.7)^2\}.$$

The parameter $b$ controls dependence strength: when $b = 0$, the null hypothesis $H_0 : X \perp Y$ holds. For each configuration $(n, d, \text{transform}, b)$, we run 1000 replications at level $\alpha = 0.05$. We consider $n = 500$ and $d \in \{2, 10\}$, with $b$ chosen from a small grid increasing in difficulty across transformations.

### 3.2. Size and power comparison

Figure 1 reports empirical rejection probabilities as functions of the signal strength $b$ for sample size $n = 500$ under four dependence structures and two marginal dimensions $d = \dim(X) = \dim(Y)$. When $b = 0$, all procedures maintain empirical sizes close to the nominal $5\%$ level in both dimensions, confirming appropriate type I error control for CoBET, dCoBET, wa-dCoBET, HSIC, dCov, RDC, Xi-mean, and MultiFIT.

In the low-dimensional setting ($d = 2$), all methods exhibit increasing power with $b$. CoBET, dCoBET, wa-dCoBET, and RDC are generally the most competitive, while Xi-mean is noticeably less powerful and MultiFIT tends to be conservative. Differences across methods remain modest for Trig, ExpQuad, and Linear transformations.

Under the LogQuad transformation, clearer separation emerges. HSIC, dCov, and Xi-mean show delayed power growth, while RDC improves but still lags behind CoBET, dCoBET, and wa-dCoBET. MultiFIT becomes competitive at moderate signals, but wa-dCoBET and CoBET consistently achieve higher power at weaker signals, highlighting the benefit of adaptive multiscale aggregation.

In higher dimension ($d = 10$), differences become more pronounced. CoBET, dCoBET, and wa-dCoBET—especially wa-dCoBET—achieve earlier detection and steeper power curves. HSIC and dCov require stronger signals, while RDC and MultiFIT lose sensitivity and Xi-mean remains underpowered.

The LogQuad case is the most challenging. wa-dCoBET and CoBET attain high power even at moderate signals, whereas HSIC, dCov, RDC, and Xi-mean lag behind. MultiFIT becomes competitive only at stronger signals but is less effective for weak dependence. This highlights the robustness of adaptive binary expansion in complex high-dimensional settings.

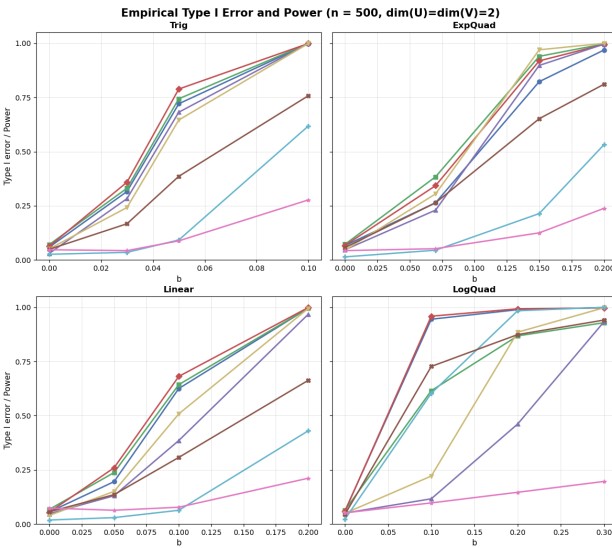

(a) $\dim(X) = \dim(Y) = 2$.

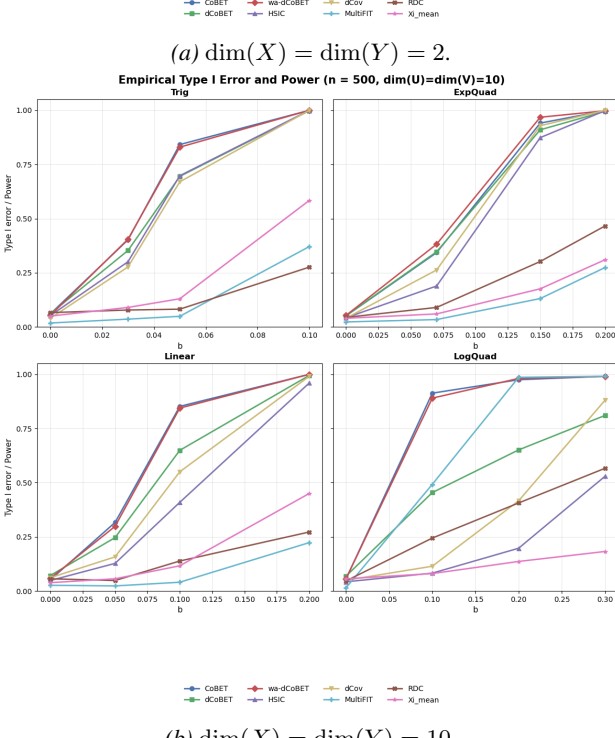

(b) $\dim(X) = \dim(Y) = 10$.

*Figure 1.* Empirical size and power comparison of CoBET, dCo-BET, wa-dCoBET, HSIC, dCov, RDC, Xi-mean, and MultiFIT as a function of $b$ for $n = 500$.

**Additional sample sizes.** Figures 5 and 6 show consistent patterns: CoBET, dCoBET, and wa-dCoBET remain strongest, RDC is competitive mainly in low dimension, MultiFIT is effective primarily for LogQuad, and Xi-mean is generally the weakest.

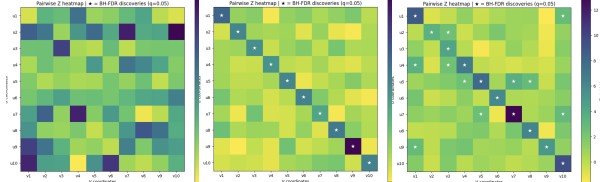

*Figure 2.* Pairwise $Z_n$–statistic heatmaps (LogQuad, $d = 10$): left: $\theta = 0$, $b = 0$; middle: $\theta = 0$, $b = 0.4$; right: $\theta = 2$, $b = 0.4$. Darker colors indicate larger $Z_n$ (proposed statistics).

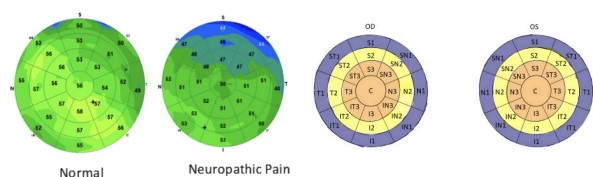

*Figure 3.* **Left:** Representative corneal epithelial thickness maps. Darker regions indicate thinner epithelium. **Right:** Regional partition used for dependence analysis. The cornea is divided into three concentric rings, with colors indicating different regions.

**Pairwise heatmap diagnostics.** To visualize coordinate-level dependence, we report heatmaps of pairwise standardized test statistics. Each cell corresponds to the $Z$-statistic for testing dependence between a coordinate of $X$ and a coordinate of $Y$, with darker colors indicating stronger evidence against independence. Multiple testing across coordinate pairs is controlled using the Benjamini–Hochberg false discovery rate (FDR) procedure (Benjamini & Hochberg, 1995); significant pairs are displayed with higher opacity.

Figure 2 illustrates three representative scenarios under the LogQuad transformation with $d = 10$. When $(\theta, b) = (0, 0)$, the heatmap shows no significant entries. When $(\theta, b) = (0, 0.4)$, dependence is restricted to matched coordinates, producing a clear diagonal pattern. When $(\theta, b) = (2, 0.4)$, dependence arises from the copula structure, yielding additional off-diagonal discoveries. These cases demonstrate how the heatmap distinguishes null behavior, sparse coordinate-wise dependence, and broader multivariate dependence.

## 4. Real-World Application

We illustrate wa-dCoBET on corneal epithelial thickness data from a dry eye disease study. The dataset contains measurements across 25 predefined corneal regions for both eyes (OD and OS) from 189 subjects, obtained via anterior segment optical coherence tomography. Each observation is represented as a spatial thickness map, with regions corresponding to angular sectors and concentric rings (See the left two in Figure 3).

### 4.1. Regional dependence analysis via wa-dCoBET

We apply wa-dCoBET to study dependence patterns in regional epithelial thickness, focusing on dependence *between* eyes across anatomically corresponding regions. To balance geometric fidelity and dimensionality, the 25 regions are grouped into three concentric sections and analyzed separately: (i) central region plus inner ring ($p = q = 10$), (ii) middle ring ($p = q = 8$), and (iii) outer ring ($p = q = 8$). See the right two in Figure 3 for an illustration. For between-eye analysis, OD and OS vectors for each concentric sections are treated as $X$ and $Y$, respectively. For example, when $(X, Y)$ are, respectively, the OD and OS epithelial thickness for the central region plus inner ring, both $X, Y$ are 10-dimensional and hence $p = q = 10$ for the central plus inner region.

We first apply the proposed procedure to detect the dependence between two vectors $X$ and $Y$ for the three concentric sections defined above. The p-values for the three sections have shown significant dependence. We set binary expansion depth $K = 5$. The adaptive weighting scheme was selected via a 10-fold (SNR) voting procedure. Multiple testing across regions was controlled using the Benjamini–Hochberg procedure at level $q = 0.05$. For the central region plus inner ring area, the proposed method produced a global test statistic of $Z = 5.749$ with p-value $4.47 \times 10^{-9}$. In comparison, HSIC test yielded a p-value of $4.99 \times 10^{-4}$ and dCov test yielded a p-value of $9.9 \times 10^{-4}$, which demonstrates the consistency of among these methods, while the proposed method provides stronger statistical evidence. To further interpret dependence, Figure 4 summarizes the inferred dependence structure. Across all three sections, wa-dCoBET detects strong *between-eye* dependence, appearing as pronounced diagonal patterns that reflect tight correspondence between matched OD and OS regions.

Between-eye dependence exhibits a clear spatial gradient. In the central section, strong off-diagonal signals indicate substantial mutual dependence among central regions, consistent with a highly coherent epithelial structure near the visual axis. This dependence weakens in the middle ring and is largely absent beyond local neighborhoods in the outer ring, indicating increasing spatial heterogeneity toward the periphery.

Overall, the analysis reveals a hierarchical organization of corneal epithelial thickness: strong bilateral symmetry across all regions, coupled with pronounced within-eye coherence in the central cornea that attenuates with distance from the center. These findings highlight the ability of wa-dCoBET to capture interpretable multiscale dependence patterns in structured biomedical data.

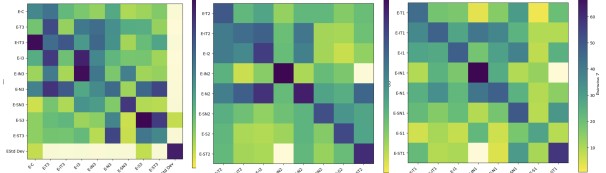

*Figure 4.* wa-dCoBET dependence heatmaps for corneal epithelial thickness regions, X-axis is the OD regions and Y-axis is the OS regions: left: Section 1 ($p = q = 10$; central + inner ring); middle: Section 2 ($p = q = 8$; middle ring); right: Section 3 ($p = q = 8$; outer ring). Strong OD–OS correspondence is visible across all sections, with Section 1 additionally exhibiting stronger within-eye coupling among central subregions.

## 5. Conclusion

We introduced a binary-expansion–based independence measure and established a general equivalence between testing independence and testing cross-covariances of binary expansion interaction coefficients for multivariate random variables. Unlike kernel-based measures such as HSIC, this equivalence holds over general spaces and does not rely on reproducing kernel Hilbert space assumptions. To address the computational challenges posed by the exponential growth of binary expansion coefficients, we developed an unbiased U-statistic and derived a scalable kernel representation that enables efficient computation. The proposed method admits a tractable asymptotic theory, eliminates the need for kernel selection, and flexibly truncates higher-order interactions to suppress noise, yielding robust power and improved interpretability.

We developed three complementary procedures: CoBET, a covariance-based test; dCoBET, a weighted extension that incorporates multiscale structure in the spirit of distance-based methods; and wa-dCoBET, an adaptive aggregation procedure that combines multiple weighting schemes via foldwise signal-to-noise-ratio voting. The resulting tests are tuning-free, computationally efficient, and admit asymptotically exact Gaussian calibration without permutation. Simulation studies demonstrate that wa-dCoBET delivers substantial power gains in nonlinear and higher-dimensional settings, while the real-data analysis of corneal epithelial thickness illustrates its interpretability and practical relevance.

### Acknowledgments

The authors appreciate the constructive feedback of the anonymous reviewers, which has significantly improved the paper. The research was partially supported by NSF grants FRG 2152289, FRG 2152070 and ECCS-2217023. Dr. William Zhao is also gratefully acknowledged for his generous support of conference-related expenses.

## Impact Statement

This work develops a new family of adaptive, distribution-free statistical tests for multivariate independence, with theoretical guarantees and efficient computational implementations. The primary impact of this research is methodological, contributing to the foundations of statistical learning and hypothesis testing. As such, we do not anticipate direct negative societal consequences arising from this work.

Potential positive impacts include improved reliability and interpretability of dependence detection in scientific applications, such as biomedical research, economics, and other data-driven domains, where understanding complex multivariate relationships is critical. By providing scalable and theoretically grounded testing procedures, this work may support more robust scientific conclusions and better-informed decision-making.

The proposed methodology is evaluated using real-world biomedical data, including human corneal epithelial thickness measurements. These data are observational in nature and are analyzed in aggregate to study statistical dependence structures, rather than to make individual-level predictions or clinical decisions. The methods do not involve intervention, diagnosis, or treatment recommendations, and any ethical considerations related to data collection, consent, or patient privacy are governed by the original data sources and institutional review processes.

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

# A. Copula-Based Data-Generating Mechanism and Simulation Results

## A.1. Simulation Results for Trigonometric to Log-Quadratic Settings with Sample Sizes $n \in \{250, 1000\}$

Figures 5 and 6 present additional empirical power curves for smaller ($n = 250$) and larger ($n = 1000$) sample sizes, respectively. These results complement the main analysis in Figure 1 and illustrate the stability of method comparisons across different sample regimes.

For $n = 250$, overall power is reduced across all procedures, particularly at small signal strengths, as expected. Nevertheless, the relative ordering of methods remains consistent with the main findings: wa-dCoBET continues to outperform competing tests under the LogQuad transformation and shows earlier detection in higher dimensions. Kernel-based methods exhibit more pronounced power degradation in complex and higher-dimensional settings.

For $n = 1000$, power increases uniformly across all alternatives, and most methods approach unit power at moderate values of $b$. Despite this, wa-dCoBET retains a clear advantage in terms of signal efficiency, achieving high power at smaller $b$ than competing approaches, especially under nonlinear and high-dimensional dependence. These patterns confirm that the improvements observed at $n = 500$ are not artifacts of a particular sample size but persist across a broad range of $n$.

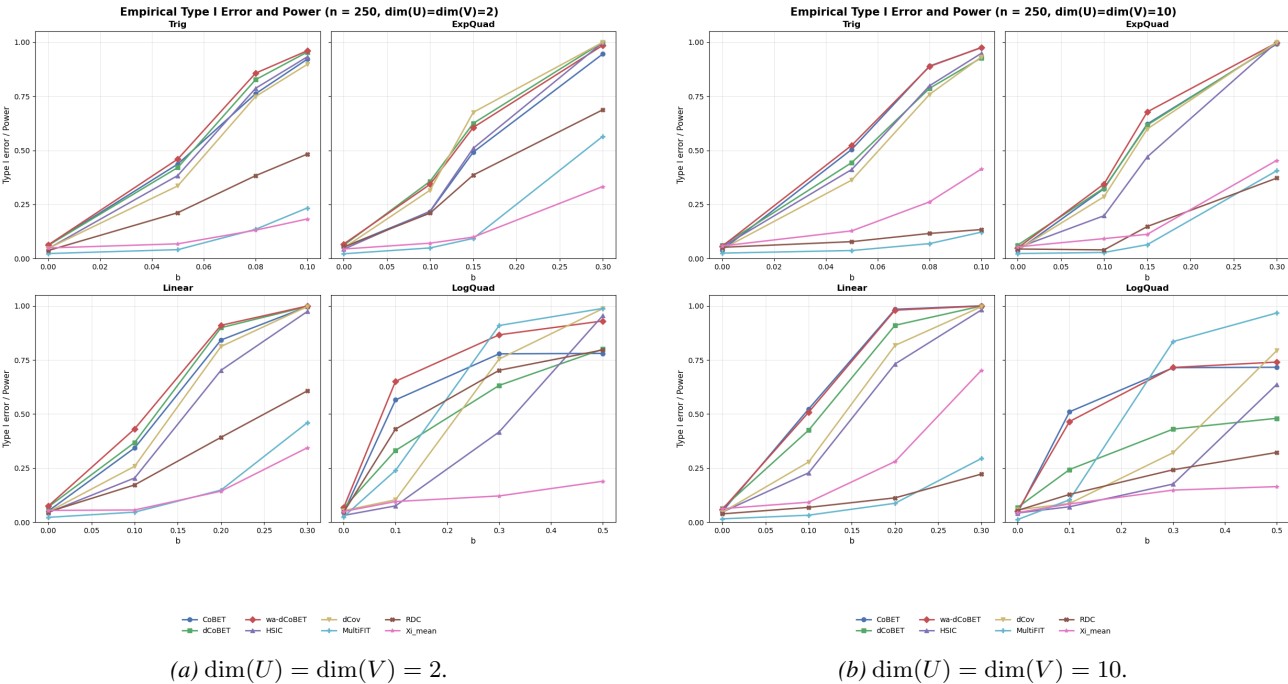

*(a)* $\dim(U) = \dim(V) = 2$.        *(b)* $\dim(U) = \dim(V) = 10$.

*Figure 5.* Empirical size and power comparison of CoBET, dCoBET, wa-dCoBET, HSIC, dCov, RDC, Xi-mean and MultiFIT as a function of signal strength $b$ for sample size $n = 250$. Results are shown for four dependence structures under low- and higher-dimensional settings. Results correspond to the same dependence structures and testing procedures as in Figure 1.

## A.2. Adaptive Weighting Behavior under Log-Quadratic Dependence

We provide results to further illustrate the behavior of the proposed adaptive weighting mechanism under complex nonlinear dependence. The proposed framework captures high-order interactions through the choice of resolution level $K$, while the SNR-based voting scheme adaptively selects weight matrices that enhance detection power.

We consider the logquad setting described in Section 3.1, which is among the most difficult scenarios due to its combination of oscillatory behavior and localized signal structure. Such dependence cannot be effectively captured by a single interaction scale and instead requires aggregating information across multiple orders. In this setting, we fix $p = q = 10$, vary the sample size $n \in \{250, 500, 1000\}$, and evaluate performance over $R = 500$ replications at significance level $\alpha = 0.05$.

The results in Table 1 show that the proposed wa-dCoBET consistently achieves power comparable to or exceeding CoBET while substantially outperforming dCoBET across all signal levels.

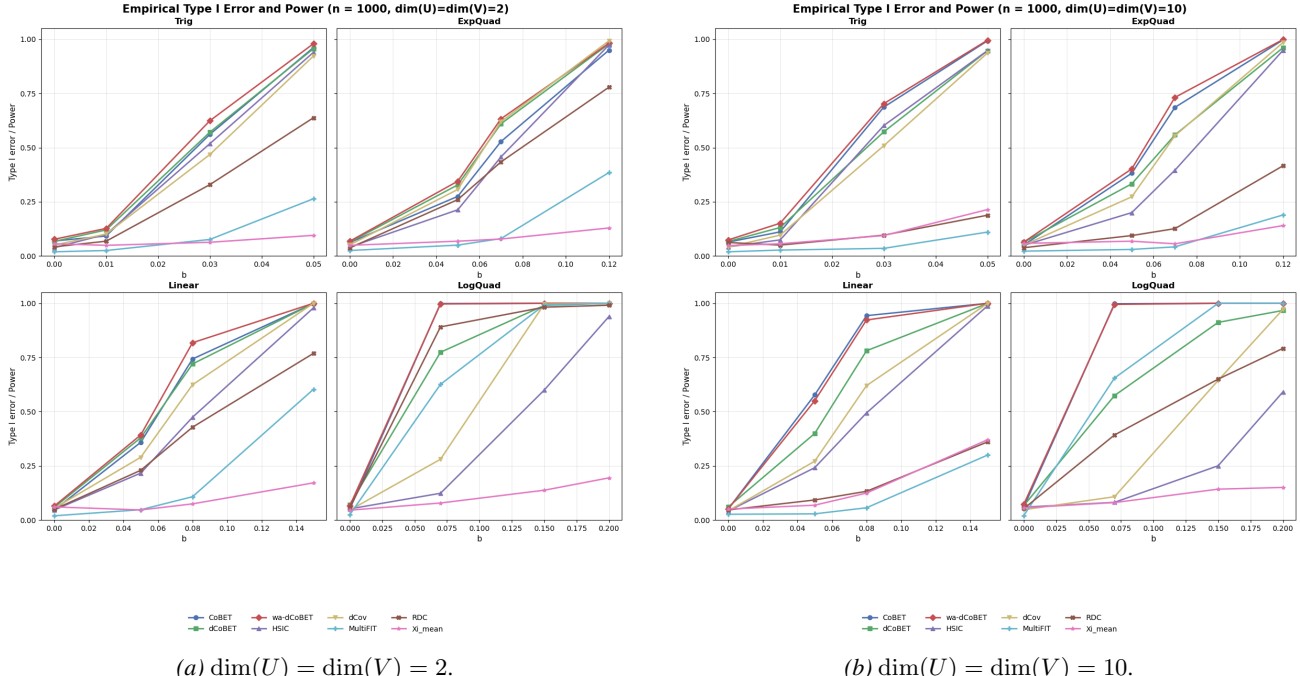

*(a)* $\dim(U) = \dim(V) = 2$.   *(b)* $\dim(U) = \dim(V) = 10$.

*Figure 6.* Empirical size and power comparison of CoBET, dCoBET, wa-dCoBET, HSIC, dCov, RDC, Xi-mean and MultiFIT as a function of signal strength $b$ for sample size $n = 1000$. Results are shown for four dependence structures under low- and higher-dimensional settings. Results correspond to the same dependence structures and testing procedures as in Figure 1.

The adaptive combination provides a clear advantage by learning weights in a data-driven approach. The learned weights reflect the relative detection powers between CoBET and dCoBET as the sample size and signal strength vary, which consistently assigns higher weights to the test with higher power. At the same time, all methods maintain controlled Type I error. Together, these results provide direct empirical evidence that the proposed adaptive weighting mechanism effectively captures complex dependence structures by dynamically balancing contributions from different interaction orders.

### A.3. Simulation Study: Comparison with HSIC with Adaptive Kernels

We conduct simulation studies to evaluate the finite-sample performance of CoBET and dCoBET, and compare them with HSIC, DCov, and the MMD-DUAL test, which uses adaptive kernel for HSIC based independence testing (Zhou et al., 2025). All methods are evaluated under the nonlinear `sinexp` setting.

Let $(U_j, V_j)$ be generated from a Clayton copula with parameter $\theta = 2$, and define

$$Z_j = \Phi^{-1}(U_j), \qquad X_j = \frac{\sin\left(\dfrac{\pi Z_j}{1 + |Z_j|}\right)}{1 + Z_j^2},$$

$$Y_j = \sin(bX_j + V_j) \exp(-bX_j^2),$$

where $\Phi^{-1}$ denotes the standard normal quantile function and $b \geq 0$ controls the signal strength. When $b = 0$, $X_j$ and $Y_j$ are independent, while increasing $b$ introduces increasingly complex nonlinear dependence.

We consider dimension $d = 5$ and sample sizes $n \in \{250, 500, 1000\}$. Signal grids are selected to capture the transition from low to high power. Each setting is evaluated using $R = 500$ Monte Carlo replications at significance level $\alpha = 0.05$.

For MMD-DUAL, the model is trained once per $(n, b)$ configuration using an independent training sample of size 500 from the joint and permuted distributions, and reused across all replications to reduce computational cost. Test thresholds are calibrated using 199 wild-bootstrap permutations.

| $n$ | $b$ | CoBET | dCoBET | wa-dCoBET | Avg $\hat{w}_{\mathrm{id}}$ | Avg $\hat{w}_D$ |
|---|---|---|---|---|---|---|
| | | | **Sample size $n = 250$** | | | |
| 250 | 0.00 | 0.045 | 0.065 | 0.051 | 0.5951 | 0.4049 |
| 250 | 0.10 | 0.510 | 0.243 | 0.465 | 0.5793 | 0.4207 |
| 250 | 0.30 | 0.714 | 0.430 | 0.715 | 0.5555 | 0.4445 |
| 250 | 0.50 | 0.716 | 0.480 | 0.720 | 0.5640 | 0.4360 |
| | | | **Sample size $n = 500$** | | | |
| 500 | 0.00 | 0.055 | 0.068 | 0.056 | 0.5629 | 0.4371 |
| 500 | 0.10 | 0.913 | 0.454 | 0.890 | 0.5898 | 0.4102 |
| 500 | 0.20 | 0.975 | 0.651 | 0.981 | 0.5541 | 0.4459 |
| 500 | 0.30 | 0.991 | 0.760 | 0.989 | 0.5528 | 0.4472 |
| | | | **Sample size $n = 1000$** | | | |
| 1000 | 0.00 | 0.059 | 0.066 | 0.072 | 0.5459 | 0.4541 |
| 1000 | 0.07 | 0.998 | 0.574 | 0.995 | 0.6383 | 0.3617 |
| 1000 | 0.15 | 1.000 | 0.912 | 1.000 | 0.6266 | 0.3734 |
| 1000 | 0.20 | 1.000 | 0.967 | 1.000 | 0.5986 | 0.4014 |

*Table 1.* Power comparison of CoBET, dCoBET, and wa-dCoBET, together with the average adaptive weights under the logquad setting ($d = 10$, $R = 500$, $\alpha = 0.05$).

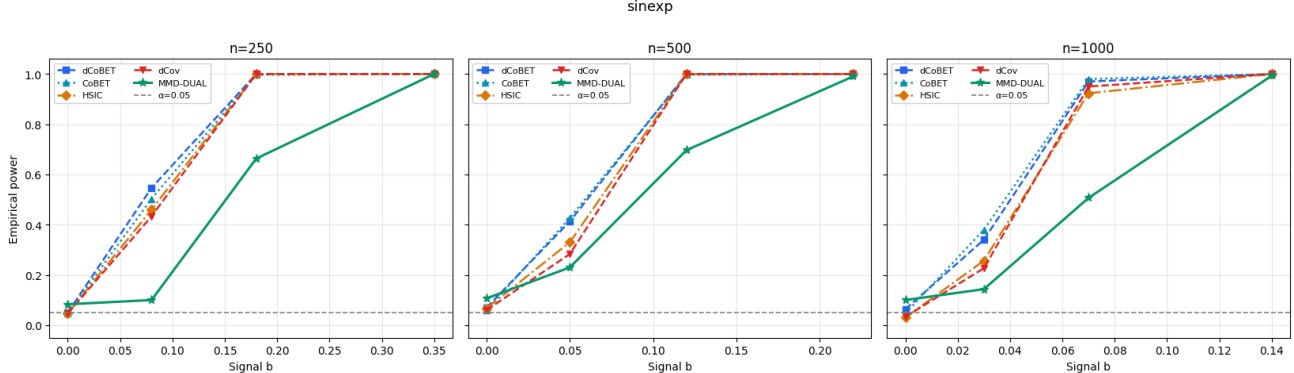

*Figure 7.* Empirical power curves for the `sinexp` transform. Clayton($\theta = 2$), $d = 5$, $\alpha = 0.05$. The dashed horizontal line marks the nominal level $\alpha = 0.05$. The leftmost point at $b = 0$ corresponds to the empirical Type I error.

Overall, dCoBET and CoBET achieve strong empirical power across all sample sizes and signal levels, particularly in weak- to-moderate signal regimes where they consistently outperform HSIC, dCov, and MMD-DUAL. The advantage is most pronounced at the weakest non-null settings, demonstrating the effectiveness of binary expansion representations for capturing localized nonlinear dependence structures under the `sinexp` alternative.

As the signal strength increases, all methods approach power close to one, indicating consistency under the alternative. In terms of Type I error control, CoBET, dCoBET, HSIC, and dCov remain close to the nominal level, while MMD-DUAL exhibits slightly inflated rejection rates in some settings together with noticeably lower power under weak signals. Overall, the results demonstrate that the proposed binary expansion-based approaches provide substantial improvements over existing kernel- and distance-based competitors in challenging nonlinear dependence settings.

### A.4. Cross-Dimensional Nonlinear Dependence

To evaluate performance of the proposed methods under more complex dependence structures, we conducted additional simulations involving *cross-dimensional interactions* and *non-separable nonlinear dependence*.

Specifically, we consider a multivariate setting with $X, Y \in \mathbb{R}^d$ ($d = 10$), where the dependence between $X$ and $Y$ is driven by interactions across multiple coordinates of $X$. Let $X$ be generated from a nonlinear transformation of a multivariate copula-based sample similar to the generation $X$ in Section 3.1. The random vector $Y = (Y_1, \cdots, Y_d)'$ is generated by the

following model:

$$Y_j = \cos\big(b \cdot (X_1 X_2 + X_3 X_4) + V\big) \cdot \exp\big(-b(X_5 - 0.5)^2\big) + \varepsilon_j, \quad j = 1, \ldots, d.$$

where $V \sim \text{Unif}(0, 1)$ and $\varepsilon_j \sim \mathcal{N}(0, \sigma^2)$ are independent noise terms. This data generation introduces several dependence and challenges: (i) multiplicative cross-dimensional interactions (e.g., $X_1 X_2$), (ii) non-additive nonlinear structure, and (iii) a shared latent signal across coordinates of $Y$ through $V$. Such dependence cannot be decomposed into coordinate-wise relationships and therefore provides a setting to evaluate the proposed method's ability to detect complex joint dependence.

We evaluate the proposed methods with sample size $n = 500$, dimension $d = 10$, resolution $K = 4$, and simulation replication $R = 500$ replications. The parameter $b$ controls the signal strength, where $b = 0$ corresponds to the null hypothesis of independence.

| $b$ | CoBET | dCoBET | wa-dCoBET | Avg $\hat{w}_{\text{id}}$ | Avg $\hat{w}_D$ |
|-----|-------|--------|-----------|--------------|-------------|
| 0.0 | 0.044 | 0.072 | 0.058 | 0.540 | 0.460 |
| 0.1 | 0.076 | 0.092 | 0.11 | 0.457 | 0.543 |
| 0.2 | 0.200 | 0.236 | 0.278 | 0.478 | 0.522 |
| 0.3 | 0.510 | 0.570 | 0.602 | 0.494 | 0.506 |
| 0.5 | 0.952 | 0.970 | 0.982 | 0.392 | 0.608 |

*Table 2*. Empirical Type I error and power under cross-dimensional nonlinear dependence.

The results demonstrate that CoBET, dCoBET and wa-dCoBET maintain valid Type I error under the null ($b = 0$), with wa-dCoBET remaining well-calibrated. Under increasing signal strength, all methods exhibit increasing power, confirming their ability to detect cross-dimensional complex dependence.

### A.5. Comparison with BET (Zhang, 2019) in the Univariate Dimension Setting

Testing independence between one-dimensional random variable, both the proposed method and Zhang's max BET (Zhang, 2019) statistic are applicable. Since Zhang's BET (Zhang, 2019) is based on the maximum norm and measures the symmetric in binary interactions, BET is more powerful when the strong dependence exists in sparse binary interactions while the proposed test is more powerful when the weak dependence exist in dense (many) binary interactions.

However, it is important to note that BET (Zhang, 2019) is not applicable to higher-dimensional case due to the computational burden since it requires to obtain the entire vector of binary expansion interactions, which grows exponentially fast as data dimension and resolution grow. The proposed method overcomes this issue by introducing the kernel representations.

To examine the relative performance of the proposed aggregation-based and max-type independence tests (Zhang, 2019), we consider the `logquad` setting introduced in Section 3.1 with $p = q = 1$. The dependence strength is controlled by the parameter $b$, where $b = 0$ corresponds to the null hypothesis and $b > 0$ introduces increasing levels of nonlinear dependence. Based on the construction, all the binary interaction coefficients are non-zero in the dependence between $X$ and $Y$, which is the dense regime.

We compare the proposed CoBET, dCoBET, wa-dCoBET, with Zhang's Max BET (Zhang, 2019) by setting the sample size at $n = 500$ and the significance level $\alpha = 0.05$. The binary expansion depth is set to $K = 4$. The results are summarized in Table 3. The results clearly illustrate the trade-off between aggregation-based and max-type statistics. Under the null

| $b$ | CoBET | dCoBET | wa-dCoBET | BET |
|-----|-------|--------|-----------|-----|
| 0.0 | 0.048 | 0.068 | 0.058 | 0.036 |
| 0.1 | 0.926 | 0.678 | 0.948 | 0.494 |
| 0.2 | 0.990 | 0.892 | 0.996 | 0.864 |
| 0.3 | 0.998 | 0.942 | 0.998 | 0.978 |
| 0.5 | 1.000 | 0.916 | 1.000 | 0.998 |

*Table 3*. Empirical size and power for the proposed tests and their comparison with BET (Zhang, 2019).

hypothesis ($b = 0$), all methods maintain valid Type I error. Under dense alternatives, however, substantial differences emerge.

In the weak-signal regime ($b = 0.1$), Max BET achieves only $0.494$ power, while wa-dCoBET reaches $0.948$, nearly doubling the detection rate. This gap highlights the well-known limitation of max-type statistics: when the signal is distributed across many components rather than concentrated in a single dominant interaction, the maximum statistic fails to accumulate evidence effectively. In contrast, wa-dCoBET aggregates information across all binary expansion coefficients through a quadratic form, enabling it to capture distributed weak signals. As the signal strength increases, the performance gap narrows, and all methods approach full power. Overall, the experiments provide empirical evidence that wa-dCoBET outperforms Max BET in dense signal regimes.

### A.6. Choosing the Binary Expansion Resolution $K$

The resolution hyperparameter $K$ may be selected in a data-driven manor by maximizing the SNR of the power function in practice. Theoretically speaking, the computational complexity of the proposed test is linear in the depth (resolution) $K$. The asymptotic power of the test is determined by the SNR estimated in the algorithm. To choose an appropriate $K$, one could find $K$ that maximizes the SNR of the test while choosing $K$ in reasonable range.

To provide practical guidance on the choice of the binary expansion depth $K$ and investigate the trade-off between power and computational efficiency, we conducted an additional sensitivity analysis of wa-dCoBET using the logquad function used simulation study (section 3.1), which is among the most challenging dependence settings considered.

In this experiment, the sample size was $n = 500$, and both $X$ and $Y$ were 10-dimensional vectors ($p = q = 10$). We evaluated several values of the binary expansion depth $K$ in the range $K \in \{3, 4, 5, 6\}$. For each $K$, we examined both type I error under the null and power under the logquad alternative with signal strength $b = 0.3$. The results are summarized in Table 4. We observe that the results were generally stable across this range. In particular, the empirical type I error remained close to the nominal $0.05$ level, while power remained high for all four choices of $K$. At the same time, the computational cost increased substantially with $K$, reflecting the exponential growth in the number of interaction terms.

These results illustrate the practical trade-off governed by $K$. Larger values of $K$ increase the resolution of the binary expansion and can capture more complex dependence structures, but they also lead to higher variance and more computational cost. Smaller values are more computational efficient, but may be less sensitive to finer-scale nonlinear dependent patterns. In our experiment, $K = 4$ achieved essentially maximal power while maintaining near-nominal type I error and substantially lower runtime than $K = 5$ or $K = 6$. Based on these considerations, we recommend selecting $K$ in a moderate range (e.g., $K = 3$–$6$) and checking robustness across nearby values, with $K = 4$ serving as a practical default for moderate sample sizes and dimensions.

| $K$ | Type I error | Power ($b = 0.3$) | Runtime (sec) |
|---|---|---|---|
| 3 | 0.064 | 0.98 | 0.026863 |
| 4 | 0.070 | 1.00 | 0.048775 |
| 5 | 0.052 | 0.97 | 0.165217 |
| 6 | 0.042 | 0.94 | 1.051935 |

*Table 4.* Sensitivity of wa-dCoBET to the binary expansion depth $K$ under the logquad setting. Reported are the empirical type I error, empirical power for the alternative with $b = 0.3$, and the average runtime (in seconds).

To evaluate computational scalability with respect to data dimension and resolution, we conducted a systematic study under the logquad function which is simulation section 3.1 by varying the sample size $n \in \{500, 1000, 1500\}$, dimension $d \in \{10, 30, 50\}$, and binary expansion depth $K \in \{3, 4, 5, 6\}$. The results are summarized in Table 5.

The empirical results show that the impact of dimension $d$ is relatively mild: increasing $d$ from 10 to 50 leads to only moderate increases in runtime, indicating that the kernel-style formulation avoids explicit enumeration of the exponentially large feature space. The expansion depth $K$ has slightly more impact on computational cost and the computational time is still reasonable. As $K$ increases, the runtime grows (e.g., from approximately 0.05 seconds at $K = 3$ to about 2 seconds at $K = 6$ when $n = 500$, $d = 50$).

Importantly, the method remains computationally efficient for moderate depths, where runtimes are well below one second even for larger sample sizes ($n = 1500$) and higher dimensions ($d = 50$). At the same time, statistical performance remains strong, with controlled Type I error and near-perfect power across all configurations. Overall, these results indicate that the proposed method scales well in practice with respect to dimension, sample size, and the choice of $K$.

| $n$ | $d$ | $K$ | Type I | Power($b = 0.2$) | Avg Runtime (sec) |
|---|---|---|---|---|---|
| 500 | 10 | 3 | 0.058 | 0.966 | 0.012 |
| 500 | 30 | 3 | 0.070 | 0.928 | 0.031 |
| 500 | 50 | 3 | 0.048 | 0.914 | 0.054 |
| 1000 | 50 | 3 | 0.082 | 1.000 | 0.094 |
| 1500 | 50 | 3 | 0.080 | 1.000 | 0.151 |
| 500 | 10 | 4 | 0.056 | 0.980 | 0.019 |
| 500 | 50 | 4 | 0.050 | 0.934 | 0.136 |
| 1000 | 50 | 4 | 0.074 | 1.000 | 0.213 |
| 1500 | 50 | 4 | 0.072 | 1.000 | 0.312 |
| 500 | 10 | 5 | 0.048 | 0.966 | 0.031 |
| 500 | 50 | 5 | 0.046 | 0.928 | 0.484 |
| 1000 | 50 | 5 | 0.082 | 1.000 | 0.697 |
| 1500 | 50 | 5 | 0.066 | 1.000 | 0.934 |
| 500 | 10 | 6 | 0.046 | 0.930 | 0.121 |
| 500 | 50 | 6 | 0.024 | 0.890 | 2.379 |
| 1000 | 50 | 6 | 0.088 | 1.000 | 3.666 |
| 1500 | 50 | 6 | 0.054 | 1.000 | 3.994 |

*Table 5.* Computational scaling of CoBET under the `logquad` setting. We vary sample size $n$, dimension $d$, and expansion depth $K$.

## B. Detailed Proof of Theorems

### B.1. Proof of Theorem 2.1

We prove the stated finite-depth expansion for $\varphi_{UV}(t, s) - \varphi_U(t)\varphi_V(s)$.

**Step 1: A single coordinate expansion.** Fix a coordinate $r \in \{1, \ldots, p\}$, a sample index $i$, and a truncation depth $K \in \mathbb{N}$. Using the truncated binary expansion

$$U_i^{(r)} = \sum_{k=1}^{\infty} 2^{-k} A_{k,i}^{(r)}, \qquad A_{k,i}^{(r)} \in \{-1, +1\},$$

we obtain

$$e^{\jmath t_r U_i^{(r)}} = \prod_{k=1}^{K} e^{\jmath t_r 2^{-k} A_{k,i}^{(r)}} = \prod_{k=1}^{K} \Big\{ \cos(t_r/2^k) + \jmath A_{k,i}^{(r)} \sin(t_r/2^k) \Big\}.$$

Expanding the product over $k$ yields a sum indexed by subsets $I_r \subset \{1, \ldots, K\}$:

$$e^{\jmath t_r U_i^{(r)}} = \sum_{I_r \subset \{1,\ldots,K\}} \jmath^{|I_r|} \Big( \prod_{k \in I_r} A_{k,i}^{(r)} \Big) \Big( \prod_{k \in I_r} \sin(t_r/2^k) \Big) \Big( \prod_{k \notin I_r} \cos(t_r/2^k) \Big).$$

Define the one-dimensional basis functions

$$\Pi_{I_r}(t_r) := \Big( \prod_{k \in I_r} \sin(t_r/2^k) \Big) \Big( \prod_{k \notin I_r} \cos(t_r/2^k) \Big), \qquad I_r \subset \{1, \ldots, K\},$$

and the associated sample-level binary interaction

$$\mathcal{A}_{i,I_r}^{(r)} := \prod_{k \in I_r} A_{k,i}^{(r)},$$

with the convention $\mathcal{A}_{i,\emptyset}^{(r)} = 1$. Then

$$e^{\jmath t_r U_i^{(r)}} = \sum_{I_r \subset \{1,\ldots,K\}} \jmath^{|I_r|} \mathcal{A}_{i,I_r}^{(r)} \Pi_{I_r}(t_r).$$

**Step 2: Multivariate factorization.** Since $e^{\jmath\langle t, U_i\rangle} = \prod_{r=1}^{p} e^{\jmath t_r U_i^{(r)}}$, multiplying the coordinatewise expansions yields

$$e^{\jmath\langle t, U_i\rangle} = \sum_{I\in\mathcal{I}_p^*} \jmath^{|I|}\mathcal{A}_{i,I}\,\Pi_I(t), \qquad |I| := \sum_{r=1}^{p}|I_r|,$$

where

$$\mathcal{A}_{i,I} := \prod_{r=1}^{p}\prod_{k\in I_r} A_{k,i}^{(r)}, \qquad \Pi_I(t) := \prod_{r=1}^{p}\Pi_{I_r}(t_r) = \prod_{r=1}^{p}\Big(\prod_{k\in I_r}\sin(t_r/2^k)\prod_{k\notin I_r}\cos(t_r/2^k)\Big).$$

Analogously, for $V_i = (V_i^{(1)}, \ldots, V_i^{(q)})$ with binary coefficients $B_{j,i}^{(s)}$, we have

$$e^{\jmath\langle s, V_i\rangle} = \sum_{J\in\mathcal{I}_q} \jmath^{|J|}\mathcal{B}_{i,J}\,\Pi_J(s), \qquad \mathcal{B}_{i,J} := \prod_{s=1}^{q}\prod_{j\in J_s} B_{j,i}^{(s)}.$$

**Step 3: Joint characteristic function expansion.** Taking expectations over the i.i.d. sample $(U_i, V_i)$,

$$\varphi_{UV}(t,s) = \mathbb{E}\Big[e^{\jmath\langle t, U_i\rangle}e^{\jmath\langle s, V_i\rangle}\Big] = \sum_{I\in\mathcal{I}_p}\sum_{J\in\mathcal{I}_q} \jmath^{|I|+|J|}\mathbb{E}(\mathcal{A}_{i,I}\mathcal{B}_{i,J})\Pi_I(t)\Pi_J(s).$$

Similarly,

$$\varphi_U(t)\varphi_V(s) = \sum_{I\in\mathcal{I}_p^*}\sum_{J\in\mathcal{I}_q^*} \jmath^{|I|+|J|}\mathbb{E}(\mathcal{A}_{i,I})\,\mathbb{E}(\mathcal{B}_{i,J})\Pi_I(t)\Pi_J(s).$$

Subtracting yields

$$\varphi_{UV}(t,s) - \varphi_U(t)\varphi_V(s) = \sum_{I\in\mathcal{I}_p^*}\sum_{J\in\mathcal{I}_q^*} \jmath^{|I|+|J|}m_{I,J}\,\Pi_I(t)\Pi_J(s),$$

where

$$m_{I,J} := \mathrm{Cov}\big(\mathcal{A}_{i,I},\mathcal{B}_{i,J}\big) = \mathbb{E}(\mathcal{A}_{i,I}\mathcal{B}_{i,J}) - \mathbb{E}(\mathcal{A}_{i,I})\,\mathbb{E}(\mathcal{B}_{i,J}).$$

Finally, if $I = (\emptyset, \ldots, \emptyset)$ or $J = (\emptyset, \ldots, \emptyset)$, then $\mathcal{A}_{i,I} \equiv 1$ or $\mathcal{B}_{i,J} \equiv 1$, implying $m_{I,J} = 0$. Hence the sum was restricted to $\mathcal{I}_p^\star$ and $\mathcal{I}_q^\star$, completing the proof. $\qquad\square$

**Remarks for Theorem 2.1.** Theorem 2.1 expresses the deviation from independence $\varphi_{UV}(t,s) - \varphi_U(t)\varphi_V(s)$ as a finite linear combination of the binary–expansion covariance coefficients

$$m_{I,J} = \mathrm{Cov}(\mathcal{A}_{i,I}, \mathcal{B}_{i,J}),$$

with basis functions $\Pi_I(t)\Pi_J(s)$. Any choice of non-negative weight matrices $(W_A, W_B)$ therefore induces a quadratic aggregation of these coefficients through (B.2). Specific selections recover familiar tests within this framework: CoBET corresponds to $W_A = I_{d_A}$ and $W_B = I_{d_B}$, while dCoBET corresponds to distance covariance-motivated weights that encode multiscale structure across the binary–expansion indices.

## B.2. Unbiased $U$–statistic estimators

Fix $K$ and let $A_i \in \mathbb{R}^{d_A}$ and $B_i \in \mathbb{R}^{d_B}$ denote the binary–expansion feature vectors constructed from $(U_i, V_i)$, indexed by $I \in \mathcal{I}_p^\star$ and $J \in \mathcal{I}_q^\star$, respectively. Let $\mu_A := \mathbb{E}(\vec{A}_i)$ and $\mu_B := \mathbb{E}(\vec{B}_i)$, and define the centered features $\tilde{A}_i := \vec{A}_i - \mu_A$ and $\tilde{B}_i := \vec{B}_i - \mu_B$. Write

$$\Sigma_{AB} := \mathrm{Cov}(\tilde{A}_i, \tilde{B}_i), \qquad \Sigma_{BA} = \Sigma_{AB}^\top.$$

For non-negative definite weight matrices $W_A \succeq 0$ and $W_B \succeq 0$, recall the weighted population functional

$$\mathcal{V}_{W,K}^2(U,V) := \mathrm{tr}\big(\Sigma_{AB}W_B\Sigma_{BA}W_A\big).$$

**Population algebra.**  Using $\Sigma_{AB} = \mathbb{E}(\vec{A}\vec{B}^\top) - \mu_A\mu_B^\top$, we expand

$$
\begin{aligned}
V_{W,K}^2(U,V) &= \mathrm{tr}\big\{\big(\mathbb{E}(\vec{A}\vec{B}^\top) - \mu_A\mu_B^\top\big)W_B\big(\mathbb{E}(\vec{B}\vec{A}^\top) - \mu_B\mu_A^\top\big)W_A\big\} \\
&= \mathrm{tr}\big\{\mathbb{E}(\vec{A}\vec{B}^\top)\,W_B\,\mathbb{E}(\vec{B}\vec{A}^\top)\,W_A\big\} - 2\,\mathrm{tr}\big\{\mathbb{E}(\vec{A}\vec{B}^\top)\,W_B\,\mu_B\mu_A^\top\,W_A\big\} \\
&\quad + \mathrm{tr}\big\{\mu_A\mu_B^\top\,W_B\,\mu_B\mu_A^\top\,W_A\big\}.
\end{aligned}
$$

**Sample analogues.**  Let $\mathbb{E}_n$ denote the empirical mean. Motivated by the three population trace terms above, define

$$
\begin{aligned}
T_{1n}^{(W)} &:= \mathrm{tr}\big\{\mathbb{E}_n(\vec{A}\vec{B}^\top)\,W_B\,\mathbb{E}_n(\vec{B}\vec{A}^\top)\,W_A\big\}, \\
T_{2n}^{(W)} &:= \mathrm{tr}\big\{\mathbb{E}_n(\vec{A}\vec{B}^\top)\,W_B\,\mathbb{E}_n(\vec{B})\,\mathbb{E}_n(\vec{A})^\top\,W_A\big\}, \\
T_{3n}^{(W)} &:= \mathrm{tr}\big\{\mathbb{E}_n(\vec{A})\,\mathbb{E}_n(\vec{B})^\top\,W_B\,\mathbb{E}_n(\vec{B})\,\mathbb{E}_n(\vec{A})^\top\,W_A\big\}.
\end{aligned}
$$

Each term admits a symmetrized $U$–statistic representation:

$$
\begin{aligned}
T_{1n}^{(W)} &= \frac{1}{n(n-1)}\sum_{i \neq j}(\vec{A}_i^\top W_A \vec{A}_j)\,(\vec{B}_j^\top W_B \vec{B}_i), \\
T_{2n}^{(W)} &= \frac{1}{n(n-1)(n-2)}\sum_{i \neq j \neq k}(\vec{A}_i^\top W_A \vec{A}_j)\,(\vec{B}_j^\top W_B \vec{B}_k), \\
T_{3n}^{(W)} &= \frac{1}{n(n-1)(n-2)(n-3)}\sum_{i \neq j \neq k \neq \ell}(\vec{A}_i^\top W_A \vec{A}_j)\,(\vec{B}_k^\top W_B \vec{B}_\ell).
\end{aligned}
$$

Finally, define the weighted CoBET/dCoBET statistic

$$
T_n^{(W)} := T_{1n}^{(W)} - 2T_{2n}^{(W)} + T_{3n}^{(W)}.
$$

**Unbiasedness.**  $T_n^{(W)}$ is an unbiased $U$–statistic estimator of $V_W^2(U,V)$.

### B.3. Computational simplifications

To reduce computational cost for $T_{2n}^{(W)}$ and $T_{3n}^{(W)}$, we exploit algebraic structure in the weighted inner products appearing in $T_{2n}^{(W)}$ and $T_{3n}^{(W)}$. Define the stacked feature vectors

$$
\vec{C}_i := \begin{pmatrix} W_A^{1/2}\vec{A}_i \\ W_B^{1/2}\vec{B}_i \end{pmatrix} \in \mathbb{R}^{d_A + d_B}, \qquad i = 1,\ldots,n.
$$

Then

$$
\vec{C}_i^\top \vec{C}_j = \vec{A}_i^\top W_A \vec{A}_j + \vec{B}_i^\top W_B \vec{B}_j.
$$

Recall the weighted $U$–statistics

$$
\begin{aligned}
T_{1n}^{(W)} &= \frac{1}{n(n-1)}\sum_{i \neq j}(\vec{A}_i^\top W_A \vec{A}_j)(\vec{B}_j^\top W_B \vec{B}_i), \\
T_{2n}^{(W)} &= \frac{1}{n(n-1)(n-2)}\sum_{i \neq j \neq k}(\vec{A}_i^\top W_A \vec{A}_j)(\vec{B}_j^\top W_B \vec{B}_k), \\
T_{3n}^{(W)} &= \frac{1}{n(n-1)(n-2)(n-3)}\sum_{i \neq j \neq k \neq \ell}(\vec{A}_i^\top W_A \vec{A}_j)(\vec{B}_k^\top W_B \vec{B}_\ell).
\end{aligned}
$$

**Simplification for $T_{2n}^{(W)}$.** Observe that

$$\vec{C}_i^\top \vec{C}_j \, \vec{C}_j^\top \vec{C}_k = \vec{A}_i^\top W_A \vec{A}_j \, \vec{A}_j^\top W_A \vec{A}_k + \vec{A}_i^\top W_A \vec{A}_j \, \vec{B}_j^\top W_B \vec{B}_k + \vec{A}_k^\top W_A \vec{A}_j \, \vec{B}_j^\top W_B \vec{B}_i + \vec{B}_i^\top W_B \vec{B}_j \, \vec{B}_j^\top W_B \vec{B}_k,$$

which implies

$$\sum_{i \neq j \neq k} \vec{A}_i^\top W_A \vec{A}_j \, \vec{B}_j^\top W_B \vec{B}_k = \frac{1}{2} \sum_{i \neq j \neq k} \left( \vec{C}_i^\top \vec{C}_j \, \vec{C}_j^\top \vec{C}_k - \vec{A}_i^\top W_A \vec{A}_j \, \vec{A}_j^\top W_A \vec{A}_k - \vec{B}_i^\top W_B \vec{B}_j \, \vec{B}_j^\top W_B \vec{B}_k \right).$$

Let $P_n^3 := n(n-1)(n-2)$. Substituting the above identity yields

$$P_n^3 T_{2n}^{(W)} = \frac{1}{2} \Bigg[ \sum_i \Big( \sum_{j \neq i} \vec{C}_i^\top \vec{C}_j \Big)^2 - \sum_{i \neq j} (\vec{C}_i^\top \vec{C}_j)^2$$

$$- \sum_i \Big( \sum_{j \neq i} \vec{A}_i^\top W_A \vec{A}_j \Big)^2 + \sum_{i \neq j} (\vec{A}_i^\top W_A \vec{A}_j)^2$$

$$- \sum_i \Big( \sum_{j \neq i} \vec{B}_i^\top W_B \vec{B}_j \Big)^2 + \sum_{i \neq j} (\vec{B}_i^\top W_B \vec{B}_j)^2 \Bigg].$$

**Simplification for $T_{3n}^{(W)}$.** Similarly, using the same decomposition,

$$\sum_{i \neq j \neq k \neq \ell} \vec{A}_i^\top W_A \vec{A}_j \, \vec{B}_k^\top W_B \vec{B}_\ell = \frac{1}{2} \sum_{i \neq j \neq k \neq \ell} \left( \vec{C}_i^\top \vec{C}_j \, \vec{C}_k^\top \vec{C}_\ell - \vec{A}_i^\top W_A \vec{A}_j \, \vec{A}_k^\top W_A \vec{A}_\ell - \vec{B}_i^\top W_B \vec{B}_j \, \vec{B}_k^\top W_B \vec{B}_\ell \right).$$

Let $P_n^4 := n(n-1)(n-2)(n-3)$. After combinatorial simplification,

$$P_n^4 T_{3n}^{(W)} = \frac{1}{2} \Bigg\{ \Big( \sum_{i \neq j} \vec{C}_i^\top \vec{C}_j \Big)^2 - 4 \Bigg[ \sum_i \Big( \sum_{j \neq i} \vec{C}_i^\top \vec{C}_j \Big)^2 - \sum_{i \neq j} (\vec{C}_i^\top \vec{C}_j)^2 \Bigg]$$

$$- \Big( \sum_{i \neq j} \vec{A}_i^\top W_A \vec{A}_j \Big)^2 + 4 \Bigg[ \sum_i \Big( \sum_{j \neq i} \vec{A}_i^\top W_A \vec{A}_j \Big)^2 - \sum_{i \neq j} (\vec{A}_i^\top W_A \vec{A}_j)^2 \Bigg]$$

$$- \Big( \sum_{i \neq j} \vec{B}_i^\top W_B \vec{B}_j \Big)^2 + 4 \Bigg[ \sum_i \Big( \sum_{j \neq i} \vec{B}_i^\top W_B \vec{B}_j \Big)^2 - \sum_{i \neq j} (\vec{B}_i^\top W_B \vec{B}_j)^2 \Bigg] \Bigg\}.$$

**Computational complexity.** Both $T_{2n}^{(W)}$ and $T_{3n}^{(W)}$ can therefore be computed using only pairwise inner products of $\{\vec{C}_i\}$, $\{\vec{A}_i\}$, and $\{\vec{B}_i\}$. This reduces the computational complexity from $O(n^3)$ and $O(n^4)$ to $O(n^2)$, making the proposed test scalable to large sample sizes.

**Closed-form expression of $K^{(p)}$.** Repeated application of product-to-sum identities yields the finite sine–series expansion

$$\Pi_I(t) = \sum_{\lambda \in \Lambda_I} c_{I,\lambda} \sin(\langle \lambda, t \rangle + \phi_{I,\lambda}), \quad \Lambda_I \subset \{\pm 2^{-1}, \ldots, \pm 2^{-K}\}^p, \quad \phi_{I,\lambda} \in \{0, \tfrac{\pi}{2}\}.$$

Substituting this expansion into the definition of $W_A(I, I')$ and using the identity

$$\frac{1}{c_p} \int_{\mathbb{R}^p} \frac{1 - \cos\langle t, x \rangle}{\|t\|^{p+1}} \, dt = \|x\|,$$

we obtain the closed-form expression

$$K_{I,I'}^{(p)} = \frac{1}{2} \sum_{\lambda \in \Lambda_I} \sum_{\lambda' \in \Lambda_{I'}} c_{I,\lambda} c_{I',\lambda'} \cos(\phi_{I,\lambda} - \phi_{I',\lambda'}) \big( \|\lambda + \lambda'\| - \|\lambda - \lambda'\| \big).$$

This representation makes $K^{(p)}$ explicit and computable for any fixed $(p, K)$.

### B.4. Proof of Theorem 2.2

Recall that $\vec{A}_i = (\mathcal{A}_{i,I})_{I \in \mathcal{I}_p^\star}$ where

$$\mathcal{A}_{i,I} = \prod_{r=1}^{p} \prod_{k \in I_r} A_{k,i}^{(r)}, \qquad I = (I_1, \dots, I_p) \in \mathcal{I}_p^\star.$$

**Step 1: factorization of $\vec{A}_i^\top \vec{A}_j$.** By definition,

$$\vec{A}_i^\top \vec{A}_j = \sum_{I \in \mathcal{I}_p^\star} \mathcal{A}_{i,I} \mathcal{A}_{j,I} = \sum_{I \in \mathcal{I}_p^\star} \prod_{r=1}^{p} \prod_{k \in I_r} \left( A_{k,i}^{(r)} A_{k,j}^{(r)} \right).$$

Let $x_k^{(r)} := A_{k,i}^{(r)} A_{k,j}^{(r)}$. Using the elementary identity

$$\sum_{S \subseteq \{1,\dots,K\}} \prod_{k \in S} x_k = \prod_{k=1}^{K} (1 + x_k),$$

we obtain, for each coordinate $r$,

$$\sum_{I_r \subseteq \{1,\dots,K\}} \prod_{k \in I_r} x_k^{(r)} = \prod_{k=1}^{K} \left( 1 + x_k^{(r)} \right).$$

Multiplying over $r = 1, \dots, p$ gives

$$\sum_{I \in \mathcal{I}_p} \prod_{r=1}^{p} \prod_{k \in I_r} x_k^{(r)} = \prod_{r=1}^{p} \prod_{k=1}^{K} \left( 1 + A_{k,i}^{(r)} A_{k,j}^{(r)} \right).$$

The term corresponding to $I = (\emptyset, \dots, \emptyset)$ equals $1$. Therefore

$$\vec{A}_i^\top \vec{A}_j = \prod_{r=1}^{p} \prod_{k=1}^{K} \left( 1 + A_{k,i}^{(r)} A_{k,j}^{(r)} \right) - 1,$$

which proves (1).

**Step 2: integral representation of $\vec{A}_i^\top W_A \vec{A}_j$.** Assume $W_A$ is defined via the Gram construction

$$W_A(I, I') = \frac{1}{c_p} \int_{\mathbb{R}^p} \Pi_I(t) \, \Pi_{I'}(t) \, w_0^2(t) \, dt, \qquad I, I' \in \mathcal{I}_p^\star,$$

where $\Pi_I(t)$ is the basis function in Theorem 2.1. Then

$$\vec{A}_i^\top W_A \vec{A}_j = \sum_{I, I' \in \mathcal{I}_p^\star} \mathcal{A}_{i,I} \, W_A(I, I') \, \mathcal{A}_{j,I'} = \frac{1}{c_p} \int_{\mathbb{R}^p} \left( \sum_{I \in \mathcal{I}_p^\star} \mathcal{A}_{i,I} \Pi_I(t) \right) \left( \sum_{I' \in \mathcal{I}_p^\star} \mathcal{A}_{j,I'} \Pi_{I'}(t) \right) w_0^2(t) \, dt.$$

Next, use the coordinatewise factorization

$$\Pi_I(t) = \prod_{r=1}^{p} \left( \prod_{k \in I_r} \sin \frac{t_r}{2^k} \prod_{k \notin I_r} \cos \frac{t_r}{2^k} \right) = C(t) \prod_{r=1}^{p} \prod_{k \in I_r} \tan \left( \frac{t_r}{2^k} \right),$$

where

$$C(t) := \prod_{r=1}^{p} \prod_{k=1}^{K} \cos \left( \frac{t_r}{2^k} \right), \qquad \pi_k^{(r)}(t) := \tan \left( \frac{t_r}{2^k} \right).$$

Thus

$$\sum_{I \in \mathcal{I}_p^\star} \mathcal{A}_{i,I} \Pi_I(t) = C(t) \sum_{I \in \mathcal{I}_p^\star} \prod_{r=1}^p \prod_{k \in I_r} \left( A_{k,i}^{(r)} \pi_k^{(r)}(t) \right).$$

Applying the same subset-sum identity as in Step 1 (and subtracting the empty multi-index term) gives

$$\sum_{I \in \mathcal{I}_p^\star} \mathcal{A}_{i,I} \Pi_I(t) = C(t) \left\{ \prod_{r=1}^p \prod_{k=1}^K \left( 1 + A_{k,i}^{(r)} \pi_k^{(r)}(t) \right) - 1 \right\}.$$

This completes the proof. $\qquad\qquad\qquad\qquad\qquad\qquad\qquad\qquad\qquad\qquad\qquad\qquad\qquad\quad$ $\square$

**Remarks for Theorem 2.2.** In particular, if $w_0^2(t) = \prod_{r=1}^p w_{0r}^2(t_r)$, the formula for $\vec{A}_i^\top W_A \vec{A}_j$ can be written as the product with marginal integrations,

$$\begin{aligned}
\vec{A}_i^\top W_A \vec{A}_j &= \prod_{r=1}^p \prod_{k=1}^K \int \left( 1 + A_{k,i}^{(r)} \pi_k^{(r)}(t_r) \right) \left( 1 + A_{k,j}^{(r)} \pi_k^{(r)}(t_r) \right) \cos^2(t_r/2^k) w_{0r}^2(t_r) dt_r \\
&\quad - \prod_{r=1}^p \prod_{k=1}^K \int \left( 1 + A_{k,i}^{(r)} \pi_k^{(r)}(t_r) \right) \cos^2(t_r/2^k) w_{0r}^2(t_r) dt_r \\
&\quad - \prod_{r=1}^p \prod_{k=1}^K \int \left( 1 + A_{k,j}^{(r)} \pi_k^{(r)}(t_r) \right) \cos^2(t_r/2^k) w_{0r}^2(t_r) dt_r \\
&\quad + \prod_{r=1}^p \prod_{k=1}^K \int \cos^2(t_r/2^k) w_{0r}^2(t_r) dt_r.
\end{aligned}$$

### B.5. Proof of Theorem 2.3

Let

$$\tilde{A}_i := \vec{A}_i - \mathbb{E}(\vec{A}_i), \qquad \tilde{B}_i := \vec{B}_i - \mathbb{E}(\vec{B}_i),$$

and define the covariance matrices

$$\Sigma_A := \operatorname{Cov}(\tilde{A}_i) = \mathbb{E}(\tilde{A}_i \tilde{A}_i^\top), \qquad \Sigma_B := \operatorname{Cov}(\tilde{B}_i) = \mathbb{E}(\tilde{B}_i \tilde{B}_i^\top).$$

Under $H_0 : U \perp V$, the sequences $\{\tilde{A}_i\}_{i \geq 1}$ and $\{\tilde{B}_i\}_{i \geq 1}$ are independent, and $\mathbb{E}(\tilde{A}_i) = 0, \mathbb{E}(\tilde{B}_i) = 0$.

**Step 1: Centering cancellation.** By the explicit substitution $\vec{A}_i = \tilde{A}_i + \mu_A$ and $\vec{B}_i = \tilde{B}_i + \mu_B$ (with $\mu_A = \mathbb{E}(\vec{A}_i)$ and $\mu_B = \mathbb{E}(\vec{B}_i)$), expanding each of $T_{1n}^{(W)}, T_{2n}^{(W)}, T_{3n}^{(W)}$ produces 16 terms. A direct coefficient bookkeeping over the index patterns $(i,j)$, $(i,j,k)$, and $(i,j,k,\ell)$ shows that in the linear combination $T_n^{(W)} := T_{1n}^{(W)} - 2T_{2n}^{(W)} + T_{3n}^{(W)}$, every term involving at least one $\mu_A$ or $\mu_B$ cancels. Hence

$$T_n^{(W)} = \tilde{T}_{1n}^{(W)} - 2\tilde{T}_{2n}^{(W)} + \tilde{T}_{3n}^{(W)},$$

where

$$\tilde{T}_{1n}^{(W)} = \frac{1}{n(n-1)} \sum_{i \neq j} \left( \tilde{A}_i^\top W_A \tilde{A}_j \right) \left( \tilde{B}_i^\top W_B \tilde{B}_j \right),$$

and $\tilde{T}_{2n}^{(W)}, \tilde{T}_{3n}^{(W)}$ are defined analogously by replacing $A$ and $B$ with $\tilde{A}$ and $\tilde{B}$ in the corresponding three- and four-index $U$–statistics. In the following Steps 2-4, we provide details for computing variances $\tilde{T}_{1n}^{(W)}$. The variances of $\tilde{T}_{2n}^{(W)}$ and $\tilde{T}_{3n}^{(W)}$ can be obtained similarly and hence the details are omitted.

**Step 2: Variance expansion for $\tilde{T}_{1n}^{(W)}$.** Define, for $i \neq j$,

$$X_{ij}^{(W)} := \left(\tilde{A}_i^\top W_A \tilde{A}_j\right)\left(\tilde{B}_i^\top W_B \tilde{B}_j\right), \qquad \tilde{T}_{1n}^{(W)} = \frac{1}{n(n-1)} \sum_{i \neq j} X_{ij}^{(W)}.$$

Then

$$\mathrm{Var}(\tilde{T}_{1n}^{(W)}) = \mathbb{E}\{(\tilde{T}_{1n}^{(W)})^2\} - \mathbb{E}^2(\tilde{T}_{1n}^{(W)}).$$

Expand the square:

$$\mathbb{E}\{(\tilde{T}_{1n}^{(W)})^2\} = \frac{1}{n^2(n-1)^2} \sum_{i \neq j} \sum_{k \neq \ell} \mathbb{E}\left[X_{ij}^{(W)} X_{k\ell}^{(W)}\right].$$

We now classify the pairs $(i, j)$ and $(k, \ell)$ by overlap pattern. There are three cases:

*(a) identical ordered pairs:* $(i, j) = (k, \ell)$. There are $n(n-1)$ such terms, contributing

$$\sum_{i \neq j} \mathbb{E}\left[(X_{ij}^{(W)})^2\right] = n(n-1)\,\mathbb{E}\left[(X_{12}^{(W)})^2\right].$$

*(b) one-index overlap: exactly one of $\{i, j\}$ equals one of $\{k, \ell\}$.* A representative term is $\mathbb{E}(X_{ij}^{(W)} X_{i\ell}^{(W)})$ with distinct $i, j, \ell$. By symmetry, all one-overlap terms reduce to either $\mathbb{E}(X_{12}^{(W)} X_{13}^{(W)})$ or $\mathbb{E}(X_{12}^{(W)} X_{32}^{(W)})$, which are equal under i.i.d. relabeling. The number of such ordered quadruples $(i, j, k, \ell)$ is $4n(n-1)(n-2)$, so the total contribution is

$$4n(n-1)(n-2)\,\mathbb{E}\left[X_{12}^{(W)} X_{13}^{(W)}\right].$$

*(c) disjoint pairs:* $\{i, j\} \cap \{k, \ell\} = \emptyset$. There are $n(n-1)(n-2)(n-3)$ such terms, each equal to $\mathbb{E}(X_{12}^{(W)} X_{34}^{(W)})$ by symmetry.

Putting the three cases together,

$$\mathbb{E}\{(\tilde{T}_{1n}^{(W)})^2\} = \frac{1}{n^2(n-1)^2}\left(n(n-1)\,\mathbb{E}[(X_{12}^{(W)})^2] + 4n(n-1)(n-2)\,\mathbb{E}[X_{12}^{(W)} X_{13}^{(W)}] + n(n-1)(n-2)(n-3)\,\mathbb{E}[X_{12}^{(W)} X_{34}^{(W)}]\right).$$

Also,

$$\mathbb{E}(\tilde{T}_{1n}^{(W)}) = \frac{1}{n(n-1)} \sum_{i \neq j} \mathbb{E}(X_{ij}^{(W)}) = \mathbb{E}(X_{12}^{(W)}),$$

hence

$$\mathbb{E}^2(\tilde{T}_{1n}^{(W)}) = \mathbb{E}^2(X_{12}^{(W)}).$$

Therefore,

$$\mathrm{Var}(\tilde{T}_{1n}^{(W)}) = \frac{1}{n^2(n-1)^2}\left(n(n-1)\,\mathbb{E}[(X_{12}^{(W)})^2] + 4n(n-1)(n-2)\,\mathbb{E}[X_{12}^{(W)} X_{13}^{(W)}]\right.$$
$$\left. + n(n-1)(n-2)(n-3)\,\mathbb{E}[X_{12}^{(W)} X_{34}^{(W)}]\right) - \mathbb{E}^2(X_{12}^{(W)}).$$

**Step 3: Simplifications under $H_0$.** Under $H_0$, $\vec{A}_1$ and $\vec{B}_1$ are independent copies and $\mathbb{E}(\tilde{A}_i) = \mathbb{E}(\tilde{B}_i) = 0$, so

$$\mathbb{E}(X_{12}^{(W)}) = \mathbb{E}(\tilde{A}_1^\top W_A \tilde{A}_2)\,\mathbb{E}(\tilde{B}_1^\top W_B \tilde{B}_2) = 0.$$

Moreover, when $\{1, 2\}$ and $\{3, 4\}$ are disjoint, $X_{12}^{(W)}$ is independent of $X_{34}^{(W)}$, hence

$$\mathbb{E}\left[X_{12}^{(W)} X_{34}^{(W)}\right] = \mathbb{E}(X_{12}^{(W)})\,\mathbb{E}(X_{34}^{(W)}) = 0$$
$$\mathbb{E}[X_{12}^{(W)} X_{13}^{(W)}] = \mathbb{E}[(\tilde{A}_1^\top W_A \tilde{A}_2)(\tilde{B}_2^\top W_B \tilde{B}_1)(\tilde{A}_1^\top W_A \tilde{A}_3)(\tilde{B}_3^\top W_B \tilde{B}_1)] = 0.$$

Thus the disjoint-pairs term vanishes, and the last subtraction also vanishes. We obtain the exact variance decomposition

$$\mathrm{Var}(\tilde{T}_{1n}^{(W)}) = \frac{1}{n(n-1)}\,\mathbb{E}[(X_{12}^{(W)})^2].$$

**Step 4: Closed-form evaluation of $\mathbb{E}[(X_{12}^{(W)})^2]$.** Under $H_0$, $\tilde{A}$ and $\tilde{B}$ are independent, so

$$\mathbb{E}[(X_{12}^{(W)})^2] = \mathbb{E}\big[(\tilde{A}_1^\top W_A \tilde{A}_2)^2\big]\,\mathbb{E}\big[(\tilde{B}_1^\top W_B \tilde{B}_2)^2\big].$$

We compute the first factor in detail:

$$(\tilde{A}_1^\top W_A \tilde{A}_2)^2 = \tilde{A}_1^\top W_A \tilde{A}_2\, \tilde{A}_2^\top W_A \tilde{A}_1 = \tilde{A}_1^\top W_A\,(\tilde{A}_2 \tilde{A}_2^\top)\,W_A \tilde{A}_1.$$

Taking expectation and conditioning on $\tilde{A}_1$ gives

$$\mathbb{E}\big[(\tilde{A}_1^\top W_A \tilde{A}_2)^2\big] = \mathbb{E}\Big[\tilde{A}_1^\top W_A\,\mathbb{E}(\tilde{A}_2 \tilde{A}_2^\top)\,W_A \tilde{A}_1\Big] = \mathbb{E}\Big[\tilde{A}_1^\top W_A \Sigma_A W_A \tilde{A}_1\Big].$$

Using $\mathbb{E}(\tilde{A}_1 \tilde{A}_1^\top) = \Sigma_A$ and cyclicity of trace,

$$\mathbb{E}\Big[\tilde{A}_1^\top W_A \Sigma_A W_A \tilde{A}_1\Big] = \mathrm{tr}\Big(W_A \Sigma_A W_A\,\mathbb{E}(\tilde{A}_1 \tilde{A}_1^\top)\Big) = \mathrm{tr}(W_A \Sigma_A W_A \Sigma_A) = \|W_A \Sigma_A\|_F^2.$$

Analogously,

$$\mathbb{E}\big[(\tilde{B}_1^\top W_B \tilde{B}_2)^2\big] = \mathrm{tr}(W_B \Sigma_B W_B \Sigma_B) = \|W_B \Sigma_B\|_F^2.$$

Therefore,

$$\mathbb{E}[(X_{12}^{(W)})^2] = \|W_A \Sigma_A\|_F^2\,\|W_B \Sigma_B\|_F^2.$$

**Step 5: Conclude the leading-order variance.** Substituting the expression for $\mathbb{E}[(X_{12}^{(W)})^2]$ yields

$$\mathrm{Var}(\tilde{T}_{1n}^{(W)}) = \frac{2}{n(n-1)}\|W_A \Sigma_A\|_F^2\,\|W_B \Sigma_B\|_F^2.$$

Similar to the above computations, under the null hypothesis $H_0$, it can be shown that

$$\mathrm{Var}(\tilde{T}_{2n}^{(W)}) = \frac{4}{n(n-1)(n-2)}\|W_A \Sigma_A\|_F^2\,\|W_B \Sigma_B\|_F^2.$$

$$\mathrm{Var}(\tilde{T}_{3n}^{(W)}) = \frac{4}{n(n-1)(n-2)(n-3)}\|W_A \Sigma_A\|_F^2\,\|W_B \Sigma_B\|_F^2.$$

In summary, we have $\mathrm{Var}(T_n^{(W)}) = \mathrm{Var}(\tilde{T}_{1n}^{(W)})\{1 + o(1)\}$ and

$$\mathrm{Var}(T_n^{(W)}) = \mathrm{Var}(\tilde{T}_{1n}^{(W)})\{1 + o(1)\} = \frac{2}{n(n-1)}\|W_A \Sigma_A\|_F^2\,\|W_B \Sigma_B\|_F^2\{1 + o(1)\},$$

which is (2).

**Remarks for Theorem 2.3.** To estimate the leading-order variance, note that

$$\|W_A \Sigma_A\|_F^2 = \mathrm{tr}(W_A \Sigma_A W_A \Sigma_A).$$

A ratio-consistent estimator can be constructed via $U$–statistics mirroring the bias-correction structure of $T_n^{(W)}$:

$$\|\widehat{W_A \Sigma_A}\|_F^2 = \frac{1}{n(n-1)}\sum_{i\neq j}(\vec{A}_i^\top W_A \vec{A}_j)^2 - \frac{2}{\binom{n}{3}}\sum_{i\neq j\neq k}(\vec{A}_i^\top W_A \vec{A}_j)(\vec{A}_j^\top W_A \vec{A}_k) + \frac{1}{\binom{n}{4}}\sum_{i\neq j\neq k\neq \ell}(\vec{A}_i^\top W_A \vec{A}_j)(\vec{A}_k^\top W_A \vec{A}_\ell).$$

An estimator of $\|W_B \Sigma_B\|_F^2$ is defined analogously. Both estimators depend only on pairwise inner products $\vec{A}_i^\top W_A \vec{A}_j$ and $\vec{B}_i^\top W_B \vec{B}_j$, which can be computed efficiently using Theorem 2.2 without explicitly expanding $\vec{A}_i$ and $\vec{B}_i$. $\qquad\square$

**B.6. Proof of Theorem 2.4**

This proof verifies that the leading term of our statistic falls into the degenerate RKHS $U$–statistic framework of He et al. (2023) and that their Condition (C3) holds under our trace assumptions. Consequently, the asymptotic normality result in He et al. (2023) applies to our standardized statistic $Z_n^{(W)}$.

### B.6.1. KERNEL REPRESENTATION OF THE LEADING TERM

Under $H_0 : U \perp V$, let

$$\tilde{A}_i := \vec{A}_i - \mathbb{E}(\vec{A}_i), \qquad \tilde{B}_i := \vec{B}_i - \mathbb{E}(\vec{B}_i), \qquad Z_i := (\tilde{A}_i, \tilde{B}_i).$$

Recall that each binary feature is a product of $\pm 1$ bits, hence every entry of $\vec{A}_i$ and $\vec{B}_i$ takes values in $\{-1, +1\}$. Therefore each entry of $\tilde{A}_i$ and $\tilde{B}_i$ is uniformly bounded in $[-1, 1]$, implying in particular that $\tilde{A}_i, \tilde{B}_i$ have finite moments of all orders and are sub-Gaussian (with constants independent of dimension).

For deterministic symmetric matrices $W_A, W_B$, define the symmetric order–2 kernel

$$K(Z_i, Z_j) := (\tilde{A}_i^\top W_A \tilde{A}_j)\,(\tilde{B}_i^\top W_B \tilde{B}_j), \qquad i \neq j,$$

and the associated $U$–statistic

$$\tilde{T}_{1n}^{(W)} = \frac{1}{n(n-1)} \sum_{i \neq j} K(Z_i, Z_j).$$

This matches the setup of He et al. (2023) with input $Z$ and kernel $K$.

### B.6.2. THE INDUCED SECOND-ORDER KERNEL $K_2$

Following He et al. (2023), define
$$K_2(z, z') := \mathbb{E}\{K(z, Z)\,K(Z, z')\}.$$

Let
$$\Sigma_A := \mathrm{Cov}(\tilde{A}), \qquad \Sigma_B := \mathrm{Cov}(\tilde{B}).$$

Under $H_0$, $\tilde{A} \perp \tilde{B}$, so for $z = (a, b)$ and $z' = (a', b')$,

$$K_2\big((a, b), (a', b')\big) = (a^\top W_A \Sigma_A W_A a')\,(b^\top W_B \Sigma_B W_B b').$$

Define also
$$Q_A := W_A \Sigma_A W_A, \qquad Q_B := W_B \Sigma_B W_B,$$

and
$$M_A := \Sigma_A^{1/2} W_A \Sigma_A^{1/2}, \qquad M_B := \Sigma_B^{1/2} W_B \Sigma_B^{1/2}.$$

Then
$$\mathrm{tr}(M_A^2) = \mathrm{tr}(W_A \Sigma_A W_A \Sigma_A) = \|W_A \Sigma_A\|_F^2, \qquad \mathrm{tr}(M_B^2) = \|W_B \Sigma_B\|_F^2.$$

### B.6.3. SPECTRAL FACTORIZATION AND THE QUANTITIES $V_2, V_4$

Let $\mathcal{K}$ be the integral operator $(\mathcal{K}f)(z) = \int K(z, z') f(z')\, dP(z')$ under $H_0$. Because $P = P_{\tilde{A}} \otimes P_{\tilde{B}}$ and

$$K\big((a, b), (a', b')\big) = K_A(a, a')\, K_B(b, b'), \qquad K_A(a, a') := a^\top W_A a', \ \ K_B(b, b') := b^\top W_B b',$$

we have the tensor-product decomposition $\mathcal{K} = \mathcal{K}_A \otimes \mathcal{K}_B$. Hence the eigenvalues of $\mathcal{K}$ are products of eigenvalues of $\mathcal{K}_A$ and $\mathcal{K}_B$, and therefore

$$V_2 := \sum_m \lambda_m^2 = \Big(\sum_r \alpha_r^2\Big)\Big(\sum_s \beta_s^2\Big), \qquad V_4 := \sum_m \lambda_m^4 = \Big(\sum_r \alpha_r^4\Big)\Big(\sum_s \beta_s^4\Big),$$

where $\{\alpha_r\}$ and $\{\beta_s\}$ are eigenvalues of $\mathcal{K}_A$ and $\mathcal{K}_B$, respectively. In our finite-dimensional linear-kernel setting, these satisfy

$$\sum_r \alpha_r^2 = \mathrm{tr}(M_A^2), \qquad \sum_r \alpha_r^4 = \mathrm{tr}(M_A^4),$$

and similarly for $B$, so

$$V_2 = \mathrm{tr}(M_A^2)\,\mathrm{tr}(M_B^2), \qquad V_4 = \mathrm{tr}(M_A^4)\,\mathrm{tr}(M_B^4).$$

B.6.4. MOMENT BOUNDS FOR BILINEAR AND QUADRATIC FORMS

We use the following standard bounds; the constants depend only on the (sub-Gaussian) moment parameters of $\tilde{A}, \tilde{B}$, which are finite and dimension-free because the coordinates are bounded (entries in $[-1, 1]$).

**Lemma B.1** (Bilinear/quadratic form moments). *Let $X, Y$ be independent, mean-zero, isotropic sub-Gaussian vectors in $\mathbb{R}^d$ (i.e., $\mathbb{E}[XX^\top] = \mathbb{E}[YY^\top] = I_d$). Then for any symmetric matrix $H$,*

$$\mathbb{E}[(X^\top H Y)^4] \leq C_1 \, tr(H^2)^2, \qquad \mathbb{E}[(X^\top H X)^2] \leq C_2\{tr(H^2) + tr(H)^2\},$$

*where $C_1, C_2 < \infty$ depend only on the sub-Gaussian norms of $X, Y$.*

B.6.5. VERIFICATION OF CONDITION (C3) OF HE ET AL. (2023)

**Step 1:** $\mathbb{E}\{K_2(Z_1, Z_2)^4\} = o(V_2^4)$. Under $H_0$,

$$K_2(Z_1, Z_2) = (\tilde{A}_1^\top Q_A \tilde{A}_2)(\tilde{B}_1^\top Q_B \tilde{B}_2),$$

and independence implies

$$\mathbb{E}\{K_2(Z_1, Z_2)^4\} = \mathbb{E}[(\tilde{A}_1^\top Q_A \tilde{A}_2)^4] \, \mathbb{E}[(\tilde{B}_1^\top Q_B \tilde{B}_2)^4].$$

Let $X_i = \Sigma_A^{-1/2} \tilde{A}_i$ and $Y_i = \Sigma_B^{-1/2} \tilde{B}_i$ so that $\mathbb{E}[X_i X_i^\top] = I$ and $\mathbb{E}[Y_i Y_i^\top] = I$. Then

$$\tilde{A}_1^\top Q_A \tilde{A}_2 = X_1^\top M_A^2 X_2, \qquad \tilde{B}_1^\top Q_B \tilde{B}_2 = Y_1^\top M_B^2 Y_2.$$

Applying Lemma B.1 with $H = M_A^2$ and $H = M_B^2$,

$$\mathbb{E}[(X_1^\top M_A^2 X_2)^4] \leq C_1 \, \text{tr}\big((M_A^2)^2\big)^2 = C_1 \, \text{tr}(M_A^4)^2,$$

and similarly $\mathbb{E}[(Y_1^\top M_B^2 Y_2)^4] \leq C_1 \, \text{tr}(M_B^4)^2$. Hence

$$\mathbb{E}\{K_2(Z_1, Z_2)^4\} \leq C_1^2 \, \text{tr}(M_A^4)^2 \, \text{tr}(M_B^4)^2.$$

Since $V_2^4 = \text{tr}(M_A^2)^4 \, \text{tr}(M_B^2)^4$ and by assumption

$$\text{tr}(M_A^4) = o(\text{tr}(M_A^2)^2), \qquad \text{tr}(M_B^4) = o(\text{tr}(M_B^2)^2),$$

we conclude

$$\mathbb{E}\{K_2(Z_1, Z_2)^4\} = o(V_2^4).$$

**Step 2:** $\mathbb{E}\{K_2(Z_1, Z_1)^2\} = o(nV_2^2)$. We have

$$K_2(Z_1, Z_1) = (\tilde{A}_1^\top Q_A \tilde{A}_1)(\tilde{B}_1^\top Q_B \tilde{B}_1),$$

so by independence

$$\mathbb{E}\{K_2(Z_1, Z_1)^2\} = \mathbb{E}[(\tilde{A}_1^\top Q_A \tilde{A}_1)^2] \, \mathbb{E}[(\tilde{B}_1^\top Q_B \tilde{B}_1)^2].$$

Using $X_1 = \Sigma_A^{-1/2} \tilde{A}_1$,

$$\tilde{A}_1^\top Q_A \tilde{A}_1 = X_1^\top M_A^2 X_1,$$

and Lemma B.1 yields

$$\mathbb{E}[(X_1^\top M_A^2 X_1)^2] \leq C_2\{\text{tr}((M_A^2)^2) + \text{tr}(M_A^2)^2\} = C_2\{\text{tr}(M_A^4) + \text{tr}(M_A^2)^2\}.$$

Since $\text{tr}(M_A^4) = o(\text{tr}(M_A^2)^2)$, the RHS is $O(\text{tr}^2(M_A^2))$; similarly, $\mathbb{E}[(\tilde{B}_1^\top Q_B \tilde{B}_1)^2] = O(\text{tr}^2(M_B^2))$. Therefore,

$$\mathbb{E}\{K_2(Z_1, Z_1)^2\} = O\big(\text{tr}^2(M_A^2) \, \text{tr}^2(M_B^2)\big) = O(V_2^2),$$

which implies

$$\frac{\mathbb{E}\{K_2(Z_1, Z_1)^2\}}{nV_2^2} = O(1/n) \to 0, \qquad \text{i.e.,} \qquad \mathbb{E}\{K_2(Z_1, Z_1)^2\} = o(nV_2^2).$$

**Conclusion.** Steps 1–2 verify Condition (C3) of He et al. (2023) for our kernel. Combining this with our ratio-consistent variance estimator $\widehat{\mathrm{Var}}(T_n^{(W)})$ and the degeneracy under $H_0$ yields

$$Z_n^{(W)} = \frac{T_n^{(W)}}{\sqrt{\widehat{\mathrm{Var}}(T_n^{(W)})}} \xrightarrow{d} \mathcal{N}(0,1),$$

which establishes Theorem 2.4. $\qquad\square$

### B.7. Worked Example: Binary–Expansion Feature Interactions

We illustrate the construction of the binary–expansion feature vector $\vec{A}_i$ and enumerate all interaction terms in a simple low–dimensional setting consistent with Section 2.1.

**Setup.** Let $p = 2$ and truncation depth $K = 2$. Let

$$U_i = (U_i^{(1)}, U_i^{(2)}) \in [-1,1]^2$$

denote the componentwise probability–integral transforms of $(X^{(1)}, X^{(2)})$.

For each coordinate $r \in \{1,2\}$, the truncated binary expansion is

$$U_i^{(r)} = 2^{-1}A_{1,i}^{(r)} + 2^{-2}A_{2,i}^{(r)}, \qquad A_{k,i}^{(r)} \in \{-1,+1\}.$$

**Binary coefficients.** At depth $K = 2$, each coordinate contributes two binary coefficients

$$A_{1,i}^{(r)}, \quad A_{2,i}^{(r)},$$

corresponding to the first and second dyadic resolution levels.

**Index sets.** For each coordinate $r$, the collection of nonempty subsets of $\{1,2\}$ is

$$\mathcal{I}^{(r)} = \big\{\{1\}, \{2\}, \{1,2\}\big\}.$$

The full multi–index set is

$$\mathcal{I}_2^\star = \big\{I = (I_1, I_2) : I_1, I_2 \in \mathcal{I}^{(r)} \cup \{\emptyset\}, \ (I_1, I_2) \neq (\emptyset, \emptyset)\big\}.$$

**Binary–expansion interactions.** For $I = (I_1, I_2) \in \mathcal{I}_2^\star$, the interaction feature is

$$\mathcal{A}_{i,I} = \prod_{r=1}^{2} \prod_{k \in I_r} A_{k,i}^{(r)}.$$

**All interaction terms.** In this example, $\vec{A}_i$ contains $2^{pK} - 1 = 15$ interaction features, which we list explicitly below.

**Main effects (single–coordinate interactions):**

$$A_{1,i}^{(1)}, \quad A_{2,i}^{(1)}, \quad A_{1,i}^{(1)}A_{2,i}^{(1)},$$
$$A_{1,i}^{(2)}, \quad A_{2,i}^{(2)}, \quad A_{1,i}^{(2)}A_{2,i}^{(2)}.$$

**Cross–coordinate interactions:**

$$A_{1,i}^{(1)}A_{1,i}^{(2)}, \quad A_{1,i}^{(1)}A_{2,i}^{(2)}, \quad A_{2,i}^{(1)}A_{1,i}^{(2)}, \quad A_{2,i}^{(1)}A_{2,i}^{(2)},$$
$$A_{1,i}^{(1)}A_{2,i}^{(1)}A_{1,i}^{(2)}, \quad A_{1,i}^{(1)}A_{2,i}^{(1)}A_{2,i}^{(2)},$$
$$A_{1,i}^{(1)}A_{1,i}^{(2)}A_{2,i}^{(2)}, \quad A_{2,i}^{(1)}A_{1,i}^{(2)}A_{2,i}^{(2)},$$
$$A_{1,i}^{(1)}A_{2,i}^{(1)}A_{1,i}^{(2)}A_{2,i}^{(2)}.$$

**Feature vector.**   Stacking all interactions yields the binary–expansion feature vector

$$\vec{A}_i = \left(\mathcal{A}_{i,I}\right)_{I \in \mathcal{I}_2^\star} \in \mathbb{R}^{15}.$$

**Interpretation.**   Each component of $\vec{A}_i$ corresponds to a multiscale interaction between binary digits of $U_i^{(1)}$ and $U_i^{(2)}$. Theorem 2.1 shows that dependence between $U$ and $V$ is fully characterized by the cross–covariances between these interaction features and their analogues for $V$.

