# OpenReview forum: "Adaptive Multiscale Binary Expansion Tests for Independence"
_ICML.cc/2026/Conference — ICML 2026 regular_

### Official Review · Reviewer_bUsA · 2026-03-10

**Soundness:** 3
**Presentation:** 3
**Significance:** 3
**Originality:** 2
**Overall Recommendation:** 4
**Confidence:** 3

**Summary:**

This paper proposes a novel framework for testing multivariate independence by leveraging binary expansion techniques. The core idea is to map continuous variables into a uniform space via probability integral transforms and then expand their joint distribution using a basis of binary interaction terms (Walsh functions). This decomposition allows the independence hypothesis to be formulated as a test on the coefficients of these binary interactions. The authors derive a U-statistic based estimator for the sum of squared coefficients, which serves as the test statistic. A key contribution is the development of a kernel trick that computes this high-dimensional statistic efficiently $O(n^2 pK)$ without explicitly enumerating the exponentially large set of interaction terms $2^{pK}$. Furthermore, the paper introduces an adaptive weighting scheme that data-adaptively combines different weight structures (e.g., uniform vs. distance-based) to enhance power across diverse dependency patterns. The authors establish the asymptotic normality of the proposed statistic under the null hypothesis, thereby enabling direct p-value calculation without the need for computationally expensive permutation tests.

**Compliance With Llm Reviewing Policy:**

Affirmed.

**Final Justification:**

The main concern is the originality and experiments. This paper serves as a form of extension for BET and indeed addresses the issue of computational complexity in high-dimensional settings, thus holding certain value.

**Key Questions For Authors:**

See weaknesses, especially the comparison with Zhang, 2019 [1]

[1] Zhang, Kai. "BET on independence." Journal of the American Statistical Association 114.528 (2019): 1620-1637.

**Limitations:**

yes

**Strengths And Weaknesses:**

Strengths:
- The most significant methodological contribution of this work is the shift from the Max-type statistic (dominant in Zhang, 2019) to an adaptive weighted sum statistic. Zhang’s original BET relies on identifying the single strongest interaction term, which is minimax optimal for sparse signals but suffers from power loss when dependencies are dense.
- The derivation of the closed-form kernel expression is elegant and practically vital. It circumvents the curse of dimensionality in computation, making the method scalable to moderate dimensions. Additionally, the establishment of asymptotic normality eliminates the need for permutation testing, offering a substantial speedup for large-scale datasets where $B \times n^2$ operations are prohibitive.
- Though complicated, the notations are still easy to follow.

Weaknesses:
- wa-dCoBET relies on an adaptive, data-driven aggregation of weights. While this improves statistical power, it arguably dilutes the direct interpretability of individual binary coefficients ($m_{I,J}$). Once the statistic is a weighted sum determined by a data-driven voting mechanism, it becomes difficult to attribute the rejection of the null hypothesis to specific, identifiable interaction terms. The authors should clarify how to recover meaningful structural insights from the aggregated adaptive statistic.
- Weighting Scheme: Does the proposed wa-dCoBET theoretically or empirically outperform Zhang's Max BET in dense or mixed-signal regimes where Max statistics are known to lose power? The paper needs to explicitly demonstrate this trade-off.
- Asymptotic Theory: The claim of avoiding permutation tests via asymptotic normality is significant. However, the rigor of this derivation under data-driven weight selection is critical. Does the paper address this?

---

> ### Author Rebuttal · Authors · 2026-03-29
>
> **Q1.** Thank you for the insightful comments. It is true that we cannot directly identify the contribution of each interaction term to the rejections from the global wa-dCoBET statistic. To identify the source of rejections and improve the interoperability of the method, we could apply an extra step by decomposing the global statistics into the sum of marginal statistics and interaction among marginal statistics if the dimensions of random vectors are not too high.
> When the dimension of random vectors is high, the decomposition may not be possible. We can rank the importance of the contribution of each dimension to the aggregated adaptive statistic by removing one dimension from the random vectors. Using this rank would provide some insights into the dependence structure, and then we can decompose the global statistics formed by the top-ranked dimensions. We include this discussion in Section 2.3.(https://www.dropbox.com/scl/fi/9olsrnrfyewf07cxwf1mp/Adaptive-Multiscale-Binary-Expansion-Tests-for-Independence.pdf?rlkey=q36g1qpvy3qs0ko4y8m9atkud&e=1&st=rvknmhzc&dl=0)
>
> **Q2.** Thank you for your comments regarding the comparison with Zhang's max BET statistic. Indeed, for testing independence between one-dimensional random variable, both the proposed method and Zhang's max BET statistic are applicable. Since Zhang's BET is based on the maximum norm and measures the symmetric in binary interactions, BET is more powerful when the strong dependence exists in sparse binary interactions while the proposed test is more powerful when the weak dependence exist in dense (many) binary interactions.
>
> However, it is important to note that BET is not applicable to higher-dimensional case due to the computational burden since it requires to obtain the entire vector of binary expansion interactions, which grows exponentially fast as data dimension and resolution grow. The proposed method overcomes this method by introducing the kernel representations.
>
> To examine the relative performance of the proposed aggregation-based and max-type independence tests \cite{zhang2019bet}, we consider the logquad setting introduced in Section~3.1 with $p=q=1$. The dependence strength is controlled by the parameter $b$, where $b=0$ corresponds to the null hypothesis and $b>0$ introduces increasing levels of nonlinear dependence. Based on the construction, all the binary interaction coefficients are non-zero in the dependence between $X$ and $Y$, which is the dense regime.
>
> We compare the proposed CoBET, dCoBET, wa-dCoBET, with Zhang's Max BET  by setting the sample size at $n=500$ and the significance level $\alpha=0.05$. The binary expansion depth is set to $K=4$. The results are summarized in below table:
> | $b$ | CoBET | dCoBET | wa-dCoBET | BET |
> |----|-------|--------|------------|-----|
> | 0.0 | 0.048 | 0.068 | 0.058 | 0.036 |
> | 0.1 | 0.926 | 0.678 | 0.948 | 0.494 |
> | 0.2 | 0.990 | 0.892 | 0.996 | 0.864 |
> | 0.3 | 0.998 | 0.942 | 0.998 | 0.978 |
> | 0.5 | 1.000 | 0.916 | 1.000 | 0.998 |
>
> **Table:** Empirical size and power for the proposed tests and their comparison with BET.
> The results clearly illustrate the trade-off between aggregation-based and max-type statistics. Under the null hypothesis ($b=0$), all methods maintain valid Type I error. Under dense alternatives, however, substantial differences emerge.
>
> In the weak-signal regime ($b=0.1$), Max BET achieves only $0.494$ power, while wa-dCoBET reaches $0.948$, nearly doubling the detection rate. This gap highlights the well-known limitation of max-type statistics: when the signal is distributed across many components rather than concentrated in a single dominant interaction, the maximum statistic fails to accumulate evidence effectively. In contrast, wa-dCoBET aggregates information across all binary expansion coefficients through a quadratic form, enabling it to capture distributed weak signals. As the signal strength increases, the performance gap narrows, and all methods approach full power. Overall, the experiments provide empirical evidence that wa-dCoBET outperforms Max BET in dense signal regimes.
>
> **Q3.** Thank you for the constructive comments. When the weight is fixed and given, the asymptotic normality derived in the paper works. For the weights that are selected by data-driven approach, more specifically by SNR voting, the weights converge to a constant as the number of folds increases (by the law of large numbers). Thus, the asymptotic normality still holds by applying the Slutsky Theorem. We added some discussion in Section 2.3 (https://www.dropbox.com/scl/fi/9olsrnrfyewf07cxwf1mp/Adaptive-Multiscale-Binary-Expansion-Tests-for-Independence.pdf?rlkey=q36g1qpvy3qs0ko4y8m9atkud&e=1&st=rvknmhzc&dl=0).

---

> > ### Author Rebuttal · Reviewer_bUsA · 2026-04-02
> >
> > Please make sure that these points are incorporated into the revised manuscript.

---

> > > ### Author Response · Authors · 2026-04-02
> > >
> > > Thank you very much for your time. We sincerely appreciate your constructive feedback, which has helped us improve the clarity and quality of our work. We are grateful for your careful review and valuable suggestions.

---

### Official Review · Reviewer_L3hy · 2026-03-11

**Soundness:** 3
**Presentation:** 3
**Significance:** 2
**Originality:** 2
**Overall Recommendation:** 4
**Confidence:** 3

**Summary:**

The paper introduces a family of adaptive, distribution-free independence tests for multivariate random vectors based on binary expansion coefficients. It establishes a theoretical equivalence between independence testing and the cross-covariances of these coefficients. To address the computational complexity of this exponential interaction structure, the tests are reformulated as U-statistics with an explicit kernel representation. By utilizing the multiscale nature of binary expansions, the framework adapts to unknown dependence structures by selectively truncating higher-order interactions to provide statistical power and interpretability. Additionally, the paper proposes an adaptive weighted aggregation procedure, termed wa-dCoBET. According to the reported simulations and real-data applications, wa-dCoBET matches or outperforms methods such as HSIC and distance covariance, particularly in high-dimensional and non-monotone settings, while maintaining accurate Type I error control.

**Compliance With Llm Reviewing Policy:**

Affirmed.

**Final Justification:**

The author has addressed most of my concerns and I have adjusted my score.

**Key Questions For Authors:**

see weakness

**Limitations:**

yes

**Strengths And Weaknesses:**

### **Strengths**
1. Strong Theoretical Foundation: The proposed method is well-supported by rigorous theoretical analysis, including the explicit derivation of the asymptotic distributions.

2. Clear Presentation: The paper is generally well-written, logically organized, and easy to follow.

### **Weaknesses**

**1. Flawed Rationale in the Motivation:**
The claim made in Line 83 (*"The inclusion of high-order coefficients may introduce more noise than signal, potentially degrading the power of the distance correlation test"*) is conceptually inaccurate. In the context of expansions, high-order coefficients typically correspond to high-frequency signals, which are essential for capturing highly complex, non-linear, or local dependencies. The fundamental reason distance correlation (dCov) loses power in certain scenarios is *not* because it includes too much noise from high-order coefficients, but rather because its intrinsic weighting scheme (e.g., $1/|t|^{p+q}$) heavily **suppresses (down-weights)** these high-frequency components. Therefore, dCov's limitation is primarily a matter of spectral/scale bias (failing to capture high-frequency signals), rather than a direct "signal vs. noise" trade-off. The authors should revise this motivation.

**2. Incomplete Empirical Evaluation (Missing Baselines):**
Given that the proposed method is an adaptive, multi-scale independence test, comparing it solely with standard HSIC and distance covariance is insufficient. To convincingly demonstrate the superiority of the proposed framework, the authors must include more recent and highly relevant baselines. Specifically, the following should be considered:
*   *Multi-scale independence tests:*
    [1] Gorsky, S., & Ma, L. (2022). Multi-scale Fisher’s independence test for multivariate dependence. *Biometrika*.
*   *Adaptive kernel/HSIC tests:*
    [2] Jitkrittum, W., et al. (2017). An adaptive test of independence with analytic kernel embeddings. *ICML*.
    [3] Ren, Y., et al. (2024). Learning adaptive kernels for statistical independence tests. *AISTATS*.
*   *Other fundamental/modern correlation coefficients:*
    [4] Chatterjee, S. (2021). A new coefficient of correlation. *JASA*.
    [5] Lopez-Paz, D., et al. (2013). The randomized dependence coefficient. *NeurIPS*.

**3. Unexplained Performance Gap and Lack of Weight Interpretability:**
*   **The gap between dCoBET and dCov:** The paper states that dCoBET utilizes a weighting scheme belonging to the same family as distance covariance (dCov). However, in the experimental section, dCoBET significantly outperforms dCov across the majority of cases. The authors need to provide a deeper analysis of this substantial discrepancy. Is this performance gap driven by specific experimental setups, or is there a fundamental algorithmic advantage in dCoBET that overcomes the limitations of dCov's weighting?
*   **Lack of empirical evidence for the learned weights:** The paper claims that the framework automatically adapts to unknown dependence structures by dynamically handling higher-order interactions. To substantiate this claim, the authors should provide detailed visualizations or analyses of the **learned weights**. Showing exactly which coefficients are activated would provide direct empirical evidence.

**4. Broken Cross-References:**
There are multiple unresolved cross-references throughout the manuscript (e.g., Line 179, Line 217, and Line 247). This formatting issue disrupts the reading flow and makes it difficult to verify the corresponding equations and theorems.

**5. Poor Figure Legibility:**
The font sizes for captions of figures are excessively small, likely due to modifications of the template?

***

---

> ### Author Rebuttal · Authors · 2026-03-29
>
> **Q1**. Thank you for the insightful comments. We apologize for the confusion and appreciate the opportunity to clarify our statement. You are correct that the weights
> $\|t\|^{q+1}$ down-weight the high-frequency components of the characteristic function.
>
> In our original statement, however, ``high-order coefficients'' referred to the higher-order terms in the binary expansion, rather than the high-frequency components in the Fourier (characteristic function) domain. In the context of the binary expansion, these higher-order terms correspond to higher-order interactions among the binary expansion coefficients.
>
> We have revised the introduction to clarify this distinction and avoid potential misunderstanding.(https://www.dropbox.com/scl/fi/9olsrnrfyewf07cxwf1mp/Adaptive-Multiscale-Binary-Expansion-Tests-for-Independence.pdf?rlkey=q36g1qpvy3qs0ko4y8m9atkud&e=1&st=rvknmhzc&dl=0).
>
> **Q2**. Thank you for your constructive and insightful comments. We appreciate you bringing these references to our attention.
> In response, we have expanded our empirical comparisons in this revision to include the randomized dependence coefficient \cite{Lopez-Paz2013}, Chatterjee’s rank-based correlation \cite{Chatterjee2021}, and the multiscale Fisher independence test \cite{Gorsky2022}, in addition to our previous benchmarks based on HSIC \cite{Gretton2005HSIC} and distance covariance \cite{Szekely2007}. The updated results are presented in Figure 1 (Section 3) and Figures 5–6 in the Appendix (link above).
> Across nearly all experimental scenarios in Section 3, we find that the proposed wa-dCoBET method either outperforms or performs comparably to these competing approaches. These results further demonstrate the effectiveness and robustness of the proposed independence test.
>
> **Q3**. a). Thank you for your insightful comment. Although dCoBET  and dCov share the similarity in the weighting function, based on the equation of $\mathcal{V}_D^2(U,V)$ in Section 2.1, we have shown that distance covariance is equivalent to a quadratic form of the binary expansion coefficients provided that the depth or resolution $K$ of the binary expansion is large enough (asymptotically infinity). However, the dCoBET method has the flexibility to truncate the binary expansion to remove
> the higher-order terms in the binary expansion, which involves the higher-order interactions among the binary expansion. We revise Section 2.2 to make this point clear.
>
> In our simulation settings in Section 3.1, the dependence under the alternatives mainly exist among the lower-order interaction terms in the binary expansions. Hence, the dCov method which includes the high-order interactions terms in the binary expansion could perform slightly worse than the
> dCoBET method.
>
> b). Thank you for the suggestions and insightful comments. The proposed method dynamically handling high-order interactions by appropriately choosing the resolution $K$. The proposed method adaptively choosing the weight function by using the strategy of the SNR voting, which automatically chooses weight matrices that provide better detection power.
>
> To assess the adaptivity of the proposed weighting mechanism under challenging nonlinear dependence, we consider the logquad setting described in Section 3.1, which is among the most difficult scenarios due to its combination of oscillatory behavior and localized signal structure. Such dependence cannot be effectively captured by a single interaction scale and instead requires aggregating information across multiple orders. In this setting, we fix $p=q=10$, vary the sample size $n \in \{250,500,1000\}$, and evaluate performance over $R=500$ replications at significance level $\alpha=0.05$.
>
> The results in Table 1 (Appendix A.4, link above) indicate that the proposed wa-dCoBET consistently achieves power comparable to or exceeding that of CoBET, while substantially outperforming dCoBET across all signal levels.
>
> The adaptive combination provides a clear advantage by learning weights in a data-driven approach. The learned weights reflect
> the relative detection powers between CoBET and dCoBET as the sample size and signal strength vary, which consistently assigns higher weights to the test with higher power. At the same time, all methods maintain controlled Type I error. Together, these results provide direct empirical evidence that the proposed adaptive weighting mechanism effectively captures complex dependence structures by dynamically balancing contributions from different interaction orders.
>
> **Q4.** Thank you for your careful reading. We have resolved the cross-reference issues in the revised version.
>
> **Q5**. Thank you for your careful reading. We are have made changes in figure captions to improve the readability.

---

> > ### Author Rebuttal · Reviewer_L3hy · 2026-04-02
> >
> > Thank you for your response. It has addressed most of my concerns.

---

> > > ### Author Response · Authors · 2026-04-03
> > >
> > > Thank you very much for your time and careful consideration of our manuscript. We sincerely appreciate your constructive feedback, which has been very helpful in improving the clarity and quality of our work.

---

### Official Review · Reviewer_Bf21 · 2026-03-13

**Soundness:** 3
**Presentation:** 3
**Significance:** 3
**Originality:** 2
**Overall Recommendation:** 4
**Confidence:** 3

**Summary:**

This paper studies independence testing for multivariate random vectors using binary expansion representations. After applying componentwise probability integral transforms, the authors express the variables through truncated binary expansions and characterize independence through cross-covariances of binary interaction coefficients, leading to several tests including CoBET, dCoBET, and the adaptive aggregation procedure wa-dCoBET. The numerical study compares the proposed methods with HSIC and distance covariance in simulations and presents an application to a corneal epithelial thickness dataset.

**Compliance With Llm Reviewing Policy:**

Affirmed.

**Final Justification:**

After considering both the paper and the rebuttal, I find that my concerns have been properly addressed, and the rebuttal has strengthened my confidence in the paper’s soundness, significance, and overall contribution.

**Key Questions For Authors:**

- Could the authors provide more guidance on how the binary expansion depth $K$ should be selected in practice? In particular, how sensitive is the performance of the proposed tests to the choice of $K$, especially in higher-dimensional settings?

- The paper shows that distance covariance can be expressed as a weighted form of the binary expansion covariance framework. Could the authors clarify what practical advantages the proposed dCoBET formulation provides compared with directly applying standard distance covariance tests?

- The numerical experiments mainly focus on coordinate-wise nonlinear transformations between variables. Could the authors discuss whether the method has been evaluated, or is expected to behave similarly, under dependence structures involving cross-dimensional interactions or more complex dependence patterns?

- While the kernel-style formulation reduces the computational burden, it would be helpful to understand how the method scales in practice as the dimension $p$ and the expansion depth $K$ increase. Could the authors comment on the computational cost and practical limits in higher-dimensional problems?

**Limitations:**

Partially. The authors provide a brief discussion of potential societal impact, including possible scientific and biomedical applications, but the discussion of limitations and possible negative consequences could be more explicit.

**Strengths And Weaknesses:**

Strengths

- **Clear theoretical framework.**
  The paper presents a coherent framework that characterizes independence through cross-covariances of binary expansion interaction coefficients, thereby translating the independence testing problem into a covariance testing problem in a transformed feature space.

- **Well-developed methodology with theoretical guarantees.**
  The proposed tests are formulated using U-statistics, and the paper provides variance characterizations and asymptotic normality results under the null hypothesis, leading to analytically calibrated tests without relying on permutation procedures.

- **Computationally scalable formulation.**
  Although the binary expansion interaction space grows exponentially with dimension and expansion depth, the paper derives kernel-style representations that allow the test statistics to be computed efficiently without explicitly constructing the full feature vectors.

- **Interpretability through a multiscale representation.**
  The binary expansion representation provides a multiscale view of dependence, which may help identify which components contribute to dependence when independence is rejected.

Weaknesses

- **The novelty relative to prior binary expansion work appears incremental.**
  The core idea of representing dependence through binary expansion coefficients builds on earlier work such as BET. The main contributions here lie in multivariate extensions, weighting schemes, and computational formulations, which may be viewed as extensions of the existing framework rather than fundamentally new ideas.

- **The diversity of simulation settings is somewhat limited.**
  The numerical experiments mainly consider coordinate-wise nonlinear transformations when generating dependence. Additional experiments involving more complex dependence structures, such as cross-dimensional interactions, would strengthen the empirical evaluation.

- **Practical guidance on hyperparameter selection is limited.**
  The method relies on a truncation depth $K$ for binary expansion, but the paper provides limited discussion of how this parameter should be selected in practice and how sensitive the method is to this choice.

- **The connection to existing methods could be clarified more explicitly.**
  While the paper highlights the relationship between the proposed framework and distance covariance, the practical advantages over directly using existing methods such as distance covariance or HSIC could be discussed more explicitly.

---

> ### Author Rebuttal · Authors · 2026-03-29
>
> **Q1.** Thank you for the insightful and constructive comments. We refer the reviewer to our response to Reviewer sbgr’s Q1, where this point is addressed in detail.
>
> **Q2.** Thank you for the insightful comments. Distance covariance can be approximated by a quadratic form of the binary expansion coefficients when the depth (or resolution) $K$ is large (i.e., theoretically $K\to\infty$). This implies that the expansion incorporates all higher-order binary coefficients, even though many of these terms may not contribute meaningfully to the dependence between two random vectors. A key advantage of dCoBET is its flexibility in selecting an appropriate
> $K$ to truncate higher-order expansions. This truncation can help reduce noise, improve the signal-to-noise ratio, and ultimately enhance the power of the proposed dCoBET test. Moreover, the proposed method allows for the incorporation of alternative weighting schemes that can be adapted to different dependence structures, further improving its flexibility and effectiveness. See the discussion in Section 2.2 (https://www.dropbox.com/scl/fi/9olsrnrfyewf07cxwf1mp/Adaptive-Multiscale-Binary-Expansion-Tests-for-Independence.pdf?rlkey=q36g1qpvy3qs0ko4y8m9atkud&e=1&st=rvknmhzc&dl=0).
>
> **Q3.** We thank the reviewer for raising this important point. To evaluate performance of the proposed methods under more complex dependence structures, we conducted additional simulations involving cross-dimensional interactions and non-separable nonlinear dependence.
>
> Specifically, we consider a multivariate setting with $X, Y \in \mathbb{R}^d$ ($d=10$), where the dependence between $X$ and $Y$ is driven by interactions across multiple coordinates of $X$. Let $X$ be generated from a nonlinear transformation of a multivariate copula-based sample similar to the generation $X$ in Section 3.1. We kindly refer the reviewer to Appendix A.5 (link above), where the detailed data generation procedure is provided.
> This data generation introduces several dependence and challenges:
> (i) multiplicative cross-dimensional interactions (e.g., $X_1 X_2$),
> (ii) non-additive nonlinear structure,
> and (iii) a shared latent signal across coordinates of $Y$ through $V$.
> Such dependence cannot be decomposed into coordinate-wise relationships and therefore provides a setting to evaluate the proposed method's ability to detect complex joint dependence.
>
> We evaluate the proposed methods with sample size $n=500$, dimension $d=10$, resolution $K=4$, and simulation replication $R=500$ replications. The parameter $b$ controls the signal strength, where $b=0$ corresponds to the null hypothesis of independence.
> | $b$ | CoBET | dCoBET | wa-dCoBET | $\mbox{Avg}\;w_{\text{id}}$ | $\mbox{Avg}\; w_{J}$ |
> |------|-------|--------|------------|------------------------------|------------------------|
> | 0.0  | 0.044 | 0.072  | 0.058      | 0.540                        | 0.460                  |
> | 0.1  | 0.076 | 0.092  | 0.11       | 0.457                        | 0.543                  |
> | 0.2  | 0.200 | 0.236  | 0.278      | 0.478                        | 0.522                  |
> | 0.3  | 0.510 | 0.570  | 0.602      | 0.494                        | 0.506                  |
> | 0.5  | 0.952 | 0.970  | 0.982      | 0.392                        | 0.608                  |
>
> **Table:** Empirical Type I error and power under cross-dimensional nonlinear dependence.
>
> The results demonstrate that CoBET, dCoBET and wa-dCoBET maintain valid Type I error under the null ($b=0$), with wa-dCoBET remaining well-calibrated. Under increasing signal strength, all methods exhibit increasing power, confirming their ability to detect cross-dimensional complex dependence.
>
> **Q4.**  Thank you for the insightful comments.
> To evaluate computational scalability with respect to data dimension and resolution, we conducted a study under the logquad function (Section 3.1), varying the sample size $n \in \{500,1000,1500\}$, dimension $d \in \{10,30,50\}$, and binary expansion depth $K \in \{3,4,5,6\}$. The results are summarized in Table 5 in Appendix A.7.
>
> The empirical results show that the impact of dimension $d$ is mild, with only moderate increases in runtime as $d$ grows. The expansion depth $K$ has a larger impact on computational cost, though runtime remains reasonable (e.g., from about 0.05 seconds at $K=3$ to about 2 seconds at $K=6$ when $n=500$, $d=50$).
>
> Importantly, the method remains computationally efficient for moderate depths, with runtimes below one second even for larger sample sizes ($n=1500$) and higher dimensions ($d=50$), while maintaining controlled Type I error and high power.
>
> Overall, these results indicate that the proposed method scales well in practice with respect to dimension, sample size, and the choice of $K$.

---

> > ### Author Rebuttal · Reviewer_Bf21 · 2026-04-04
> >
> > Thank you for your detailed response. My questions have been adequately addressed. I will maintain my original score, which is already on the positive (accept) side.

---

> > > ### Author Response · Authors · 2026-04-04
> > >
> > > Thank you very much for your time and careful consideration of our manuscript. We sincerely appreciate your constructive feedback, which has been very helpful in improving the clarity and quality of our work.

---

### Official Review · Reviewer_sbgr · 2026-03-13

**Soundness:** 3
**Presentation:** 3
**Significance:** 3
**Originality:** 3
**Overall Recommendation:** 5
**Confidence:** 3

**Summary:**

This paper introduces a new family of adaptive, distribution-free independence tests for multivariate random vectors based on binary expansion coefficients, supported by rigorous asymptotic theory. The core theoretical contribution establishes an equivalence between statistical independence and the vanishing of cross-covariances among exponentially many binary interaction coefficients—a characterization that extends beyond kernel-based RKHS representations and avoids manual kernel selection. To overcome the computational challenge posed by the exponential number of interactions, the authors reformulate the tests as U-statistics and derive explicit kernel representations that reduce complexity from \(O(2^{dK})\) to \(O(dK)\), enabling scalable implementation. The framework also enhances interpretability by identifying which binary expansion coefficients drive rejection of the null hypothesis.

Building on this foundation, the paper develops three test variants: CoBET (a baseline with identity weights), dCoBET (incorporating distance-covariance-inspired weights), and wa-dCoBET (an adaptive aggregation that combines both weighting schemes via foldwise signal-to-noise ratio voting). Extensive simulations and a real-data application to corneal epithelial thickness measurements demonstrate that wa-dCoBET consistently matches or outperforms competing methods (including HSIC and distance covariance) in detection power across linear and nonlinear dependence structures, particularly in high-dimensional settings, while maintaining accurate Type I error control. The proposed tests are theoretically transparent, computationally efficient, and robust to diverse dependence patterns, making them well-suited for modern machine learning and statistical applications.

**Compliance With Llm Reviewing Policy:**

Affirmed.

**Key Questions For Authors:**

Q1: The binary expansion depth K is a key parameter. Could you provide practical guidance on selecting K for real-world datasets (e.g., heuristics, cross-validation, or bounds based on sample size n and dimension d)? Additionally, how does K affect the trade-off between power and computational efficiency across different dependence structures?

Q2: The real-data application in Section 4 would benefit from more quantitative detail. Could you include comparisons with baseline methods (e.g., HSIC, distance covariance) and report key statistics such as p-values or rejection rates? Clarifying preprocessing steps and hyperparameter settings (e.g., number of folds for SNR voting) would also improve reproducibility.

**Limitations:**

No. The paper does not discuss the sensitivity to the choice of binary expansion depth K, the lack of guidance on selecting K in practice, or how the method performs under dependence structures not covered by the simulations.

**Strengths And Weaknesses:**

This paper makes several interesting contributions to independence testing. The theoretical core establishes an equivalence between statistical independence and cross-covariances of binary expansion coefficients, which offers a fresh perspective beyond RKHS-based methods and avoids the need for manual kernel selection. The computational advance is particularly clever: by reformulating the tests as U-statistics and deriving kernel representations, the authors reduce complexity from exponential \(O(2^{dK})\) to linear \(O(dK)\) in dimension and expansion depth, making the method feasible for higher-dimensional data. The wa-dCoBET test adaptively combines different weighting schemes via signal-to-noise ratio voting, and simulations suggest it performs well across linear and nonlinear dependence structures—especially in higher dimensions—while maintaining Type I error control through asymptotic approximations that avoid permutation. The framework also offers some interpretability by identifying which binary expansion coefficients contribute to rejection, which could be useful in practice. Overall, the paper presents a well-thought-out methodology with solid theoretical support.

A few aspects could be clarified or strengthened. The binary expansion depth 𝐾 is an important tuning parameter, and the paper does not provide much guidance on how to choose it in practice; it would be helpful to know what values were used in the experiments and whether results are sensitive to this choice. The real-data application in Section 4 feels a bit brief—it would benefit from more detail on the data, the preprocessing steps, and quantitative comparisons with baseline methods like HSIC or distance covariance, rather than relying mostly on visual inspection. The theoretical analysis and simulations focus on dependence structures generated by Clayton copulas and specific transformations; it is not entirely clear how the tests would behave in other challenging scenarios (e.g., highly noisy or sparse dependence). Additionally, while the asymptotic results are reassuring, a more thorough investigation of finite-sample behavior—particularly in very high dimensions or with limited sample sizes—would help readers understand when the method can be trusted. These are more about clarity and completeness than fundamental flaws, and addressing them would make the paper even stronger.

---

> ### Author Rebuttal · Authors · 2026-03-29
>
> **Q1.** Thank you for the insightful comments. The resolution hyperparameter K may be selected in a data-driven manor by maximizing the SNR of the power function in practice.
>
> Theoretically speaking, the computational complexity of the proposed test is linear in the depth (resolution) K. The asymptotic power of the test is determined by the SNR estimated in the algorithm. To choose an appropriate K, one could find K that maximizes the SNR of the test while choosing K in a reasonable range.
>
> To provide practical guidance on the choice of the binary expansion depth K and investigate the trade-off between power and computational efficiency, we conducted an additional sensitivity analysis of wa-dCoBET using the logquad function used simulation study (Section 3.1), which is among the most challenging dependence settings considered.
>
> In this experiment, the sample size was n=500, and both X and Y were 10-dimensional vectors (p=q=10). We evaluated several values of the binary expansion depth K in the range \(K in \{3,4,5,6\}\). For each K, we examined both type I error under the null and power under the logquad alternative with signal strength \(b=0.3\). The results are summarized below.
>
> | K | Type I error | Power (b=0.3) | Runtime (sec) |
> |------|-------------|------------------|---------------|
> | 3 | 0.064 | 0.98 | 0.026863 |
> | 4 | 0.070 | 1.00 | 0.048775 |
> | 5 | 0.052 | 0.97 | 0.165217 |
> | 6 | 0.042 | 0.94 | 1.051935 |
>
> We observe that the results were generally stable across this range. In particular, the empirical type~I error remained close to the nominal \(0.05\) level, while power remained high for all four choices of K. At the same time, the computational cost increased substantially with K, reflecting the exponential growth in the number of interaction terms.
>
> These results illustrate the practical trade-off governed by K. Larger values of K increase the resolution of the binary expansion and can capture more complex dependence structures, but they also lead to higher variance and more computational cost. Smaller values are more computational efficient, but may be less sensitive to finer-scale nonlinear dependent patterns.
>
> In our experiment, K=4 achieved essentially maximal power while maintaining near-nominal type~I error and substantially lower runtime than K=5 or K=6. Based on these considerations, we recommend selecting K in a moderate range (e.g., K=3--6) and checking robustness across nearby values, with K=4 serving as a practical default for moderate sample sizes and dimensions.
>
> **Q2.** Thank you for great suggestions.  We have included
>   more details of the proposed method and its
>   comparison with the HSIC and dCov methods in Section 4.1.(https://www.dropbox.com/scl/fi/9olsrnrfyewf07cxwf1mp/Adaptive-Multiscale-Binary-Expansion-Tests-for-Independence.pdf?rlkey=q36g1qpvy3qs0ko4y8m9atkud&st=rvknmhzc&dl=0)
>
>
> For reproducibility, we restricted the analysis to subjects with both OD and OS measurements and aligned paired observations by subject ID. All corneal epithelial thickness measurements are numerical. Subjects with missing values in any of the selected features were removed, resulting in a complete-case dataset.
>
> To balance geometric fidelity and dimensionality, the 25 corneal regions were grouped into three concentric sections and analyzed separately: (i) the central region plus inner ring ($p=q=10$), (ii) the middle ring ($p=q=8$), and (iii) the outer ring ($p=q=8$). In the between-eye analysis, OD and OS measurements within each section were treated as paired random vectors $(X, Y)$. For example, for the central region plus inner ring, both $X$ and $Y$ are 10-dimensional vectors.
>
> For the proposed method, the binary expansion depth was set to $K=5$. The adaptive weighting scheme was selected via a 10-fold signal-to-noise ratio (SNR) voting procedure. Multiple testing across regions was controlled using the Benjamini--Hochberg procedure at level $q=0.05$.
>
> To benchmark performance, we applied HSIC AND dCov used dependence measures.
> For illustration, we report results for the central region plus inner ring. The proposed method produced a global test statistic of $Z = 5.749$ with p-value $4.47 \times 10^{-9}$. In comparison, HSIC yielded a p-value of $4.99 \times 10^{-4}$ and distance covariance yielded a p-value of $9.9 \times 10^{-4}$. These results demonstrate that all methods detect significant dependence between OD and OS measurements, while the proposed method provides stronger statistical evidence.

---

> > ### Author Rebuttal · Reviewer_sbgr · 2026-04-03
> >
> > I have read the authors' rebuttal and accept their responses.
> > I maintain my original score.

---

> > > ### Author Response · Authors · 2026-04-03
> > >
> > > Thank you very much for your time and careful consideration of our manuscript. We sincerely appreciate your constructive feedback, which has been very helpful in improving the clarity and quality of our work.

---

### Decision · Program_Chairs · 2026-04-30

**Decision:**

Accept (regular)

**Comment:**

This paper gives an interesting new test for independence, with asymptotic level control and an aggregation-based version for increased power. While I did not find myself totally convinced in the superiority of this method over prior methods, it is different and interesting enough that all reviewers agreed it should be presented at ICML.

Please integrate the reviewer feedback and your rebuttal comments into the final version. In addition to the changes you've already made, it would be better for fairness to also compare to optimized or aggregated variants of HSIC such as HSICAgg; in moderately high dimensions I would not be surprised if those were on par with your approach.